# Anthropogenic $CO_2$ monitoring satellite mission: the need for multi-angle polarimetric observations

Stephanie P. Rusli[1], Otto Hasekamp[1], Joost aan de Brugh[1], Guangliang Fu[1], Yasjka Meijer[2], and Jochen Landgraf[1]

[1]SRON Netherlands Institute for Space Research, Sorbonnelaan 2, 3584 CA Utrecht, The Netherlands
[2]European Space Agency, Keplerlaan 1, 2201 AZ Noordwijk, The Netherlands

**Abstract.** Atmospheric aerosols have been known to be a major source of uncertainties in $CO_2$ concentrations retrieved from space. In this study, we investigate the added value of multi-angle polarimeter (MAP) measurements in the context of the Copernicus candidate mission for anthropogenic $CO_2$ monitoring (CO2M). To this end, we compare aerosol-induced $XCO_2$ errors from standard retrievals using spectrometer only (without MAP) with those from retrievals using both MAP and spectrometer. MAP observations are expected to provide information about aerosols that is useful for improving $XCO_2$ accuracy. For the purpose of this work, we generate synthetic measurements for different atmospheric and geophysical scenes over land, based on which $XCO_2$ retrieval errors are assessed. We show that the standard $XCO_2$ retrieval approach that makes no use of auxiliary aerosol observations returns $XCO_2$ errors with an overall bias of 1.12 ppm, and a spread (defined as half of the 15.9th to the 84.1th percentile range) of 2.07 ppm. The latter is far higher than the required $XCO_2$ accuracy (0.5 ppm) and precision (0.7 ppm) of the CO2M mission. Moreover, these $XCO_2$ errors exhibit a significantly larger bias and scatter at high aerosol optical depth, high aerosol altitude, and low solar zenith angle, which could lead to a worse performance in retrieving $XCO_2$ from polluted areas where $CO_2$ and aerosols are co-emitted. We proceed to determine MAP instrument specifications in terms of wavelength range, number of viewing angles, and measurement uncertainties that are required to achieve $XCO_2$ accuracy and precision targets of the mission. Two different MAP instrument concepts are considered in this analysis. We find that for either concept, MAP measurement uncertainties on radiance and degree of linear polarization should be no more than 3% and 0.003, respectively. ~~Adopting the derived MAP requirements, a retrieval exercise using both MAP and spectrometer measurements of the synthetic scenes delivers $XCO_2$ errors with an overall bias of -0.004 ppm and a spread of 0.54 ppm, implying compliance~~ *A retrieval exercise using MAP and spectrometer measurements of the synthetic scenes is carried out for each of the two MAP concepts. The resulting XCO2 errors have an overall bias of -0.004 ppm and a spread of 0.54 ppm for one concept, and a bias of 0.02 ppm and a spread of 0.52 ppm for the other concept. Both are compliant* with the CO2M mission requirements; the very low bias is especially important for proper emission estimates. For the test ensemble, we find effectively no dependence of the $XCO_2$ errors on aerosol optical depth, altitude of the aerosol layer, and solar zenith angle. These results indicate a major improvement in the retrieved $XCO_2$ accuracy with respect to the standard retrieval approach, which could lead to a higher data yield, better global coverage, and a more comprehensive determination of $CO_2$ sinks and sources. As such, this outcome underlines the contribution of, and therefore the need for, a MAP instrument onboard the CO2M mission.

# 1 Introduction

Carbon dioxide is the most important greenhouse gas in our atmosphere. It accounts for 76% of the total anthropogenic green-house gas emissions in 2010, according to the latest assessment report (2014) of IPCC (Intergovernmental Panel on Climate Change). In an international effort to mitigate climate change, 195 countries signed the Paris agreement (United Nations Frame-work Convention on Climate Change, 2015) that aims to limit global temperature rise to less than 2 degrees Celcius above the pre-industrial levels by reducing greenhouse gas emissions. To achieve this goal, quantifications of $CO_2$ emissions on a national scale with adequate temporal and spatial resolution, and a global coverage are necessary for the implementation and evaluation of carbon reduction policies (Ciais et al., 2014). Space-based observations have the capacity to perform such task and will therefore play an important role in complementing and reinforcing $CO_2$ inventories (Ciais et al., 2015; Pinty et al., 2017). For this reason, the European Commission and the European Space Agency (ESA) proposed the Anthropogenic Carbon Dioxide Monitoring (CO2M) satellite mission as a part of a larger-scale $CO_2$ initiative within Europe's Earth observation programme Copernicus to monitor and verify man-made $CO_2$ emissions and their trends (Pinty et al., 2017).

The CO2M mission is designed as a constellation of up to 3 satellites with imaging capabilities ~~and a revisit time of 2-3 days for latitudes (poleward of) 40 degrees~~ *, providing a global coverage with a revisit time of 5 days*. Each satellite carries a primary sounder, that is a nadir-looking spectrometer that will deliver measurements of column-averaged dry-air mole fraction of carbon dioxide $XCO_2$ , defined as the ratio of the total column of $CO_2$ to that of dry air. As opposed to currently operational $CO_2$ missions that are designed to observe natural $CO_2$ fluxes, with the exception of OCO-3 (Basilio et al., 2019), the CO2M mission is intended to measure anthropogenic emissions (Pinty et al., 2017). With fossil $CO_2$ emissions primarily concentrated in urban areas, industrial sites, and point sources such as power plants and refineries, it is important to detect and localize these hotspots. Auxiliary measurements of $NO_2$ that are co-emitted with $CO_2$ plumes are proposed to help the mission distinguish anthropogenic from biospheric $CO_2$ signals (Kuhlmann et al., 2019). To further resolve and quantify these emissions, Ciais et al. (2015); Crisp et al. (2018) suggest that $XCO_2$ images should have a spatial resolution of about 4 km$^2$, an $XCO_2$ precision $\leq 0.7$ ppm, with $XCO_2$ systematic errors of 0.5 ppm or less (Meijer et al., 2019). Such $XCO_2$ accuracy requirement becomes a challenge when aerosols and thin clouds are not properly taken into account in the retrieval.

Scattering by aerosols and cirrus has long been identified as one of the main sources of uncertainties in retrieving $XCO_2$ from solar backscattered radiation (e.g. Kuang et al. (2002); Houweling et al. (2005); *Frankenberg et al. (2012)*; Jung et al. (2016)). The presence of aerosols can shorten or ~~enhance~~ *lengthen* the light path, depending on the altitude of the aerosol layer and on the reflection properties of the underlying surface. In effect, this alters the depth of $CO_2$ absorption features in that they appear shallower/deeper, which can be falsely interpreted as lower/higher atmospheric $CO_2$ concentration. Depending on the observed atmospheric scene and surface albedo, neglecting scattering in the retrieval can lead to substantial $XCO_2$ errors, which are often higher than 1% (about 4 ppm) (Aben et al., 2007; Butz et al., 2009). Methods to correct or compensate for the light path modification typically include explicit parametrization of aerosols and clouds in the $XCO_2$ retrieval as proxies to the actual scattering (e.g. Kuang et al. (2002); Oshchepkov et al. (2008); Butz et al. (2011); Morino et al. (2011); Reuter et al. (2017); Nelson and O'Dell (2019)). This approach usually involves adding one or more types of scattering particles in the forward

model and retrieving their properties while using data limited to radiometric observations. The resulting $XCO_2$ uncertainties are in most cases still larger than about 1 ppm (e.g. Butz et al. (2009))~~, which is above the CO2M error requirements.~~ *Frankenberg et al. (2012) use multi-angle measurements of high spectral-resolution radiances to decrease $XCO_2$ uncertainties due to aerosol interference and demonstrate that the errors on the retrieved $CO_2$ column could be reduced down to about 1 ppm. Despite the advancement introduced by the various methods, the $XCO_2$ uncertainties are still above the CO2M error requirements.*

In this paper, we explore the potential of having auxiliary aerosol-dedicated measurements alongside the CO2M spectrometer measurements to help achieve the required $XCO_2$ accuracy. Here, we utilize the capability of a Multi-Angle Polarimeter (MAP), which measures radiance and degree of linear polarization (DLP) simultaneously at multiple wavelengths and at multiple viewing angles. The proper interpretation of such observations are currently considered the most advanced aerosol remote sensing approach and provide the most comprehensive information about aerosol properties (Dubovik et al., 2019). There is a large variety of orbital MAP instruments, which can be broadly classified into two instrument concepts considered for the CO2M mission, referred to here as MAP-mod and MAP-band concepts. A MAP-mod instrument employs spectral polarization modulation technique such that the polarization information is encoded in the modulation pattern of the radiance spectrum (Snik et al., 2009). On the other hand, a bandpass polarimeter (MAP-band) measures radiance and polarization at specific spectral channels. Most MAP instruments fall into the MAP-band category, including the series of POLDER instruments (Deschamps et al., 1994; Tanré et al., 2011), the future 3MI instrument (Fougnie et al., 2018) on EUMETSAT Polar System - Second Generation, Multi-Angle Imager for Aerosols (MAIA) (Diner et al., 2018), and Hyper-Angular Rainbow Polarimeter-2 (HARP2) (Martins et al., 2018). ~~Linear error analysis to derive the optimal instrument specification for each of the two MAP concepts with regard to wavelength range, number of viewing angles and the measurement uncertainties are a part of our study here.~~ *Linear error analysis is part of our study, to derive the optimal instrument specification for each of the two MAP concepts with regard to wavelength range, number of viewing angles and the measurement uncertainties.* We investigate the added value of a MAP instrument as part of the CO2M mission, by comparing aerosol-induced $XCO_2$ errors from retrievals using spectrometer data only with the errors from retrievals using the combined spectrometer and MAP measurements. ~~For the retrieval input, we generate synthetic measurements that correspond to an ensemble of atmospheric and geophysical scenes over land. The MAP instrument for which the synthetic measurements are generated, is tailored to the CO2M mission precision and accuracy requirements.~~ *For the retrieval input we generate synthetic measurements that correspond to an ensemble of atmospheric and geophysical scenes over land. The MAP instrument for which the synthetic measurements are generated is tailored to the CO2M mission precision and accuracy requirements.*

In the next section, we present the generic instrument description of the spectrometer and the two MAP instrument concepts used in this study. Section 3 details our three approaches to evaluate aerosol-induced $XCO_2$ errors, i.e. a joint retrieval method that enables a synergistic use of MAP and spectrometer measurements, a linear error analysis which is employed to derive MAP instrument requirements, and a spectrometer-only retrieval method which is applied to a standard $XCO_2$ retrieval without the auxiliary MAP observations. Section 4 describes the ensemble of 500 scenes for which synthetic measurements are generated; these are used in the retrieval exercises that follow. In Section 5, we perform $XCO_2$ retrievals using only spectrometer measurements and present the results. Section 6 is dedicated to the MAP requirement study in which we apply the linear error analysis

to determine the baseline setup for two MAP instrument concepts. In section 7, we adopt one of the baseline MAP setups and implement $XCO_2$ retrievals assuming both MAP and spectrometer measurements are available. The retrieved $XCO_2$ are assessed in the same way as in the spectrometer-only approach and the comparison between the resulting $XCO_2$ uncertainties is discussed here. The final section summarizes the paper. *The main body of this paper focuses on $XCO_2$ retrievals and we place the discussion on the retrieved aerosol properties in Appendix B.*

## 2  Instruments

For the CO2M mission, a 3-band spectrometer is envisaged to be the main instrument that provides measurements necessary for the $XCO_2$ retrieval. The 3 bands comprise a NIR band at 765 nm and two shortwave infrared bands at 1.6 $\mu$m (SWIR1) and 2.0 $\mu$m (SWIR2). The NIR band, as well as the strong $CO_2$-absorption band in the SWIR2 contain information about aerosols and cirrus. We adopt here the spectrometer spectral properties as proposed for the CO2M mission, given in Table 1. Noise on radiance $I$ is calculated from $\mathrm{SNR} = a_{\mathrm{noise}} I / \sqrt{(a_{\mathrm{noise}} I + b_{\mathrm{noise}})}$, in which $a_{\mathrm{noise}}$ and $b_{\mathrm{noise}}$ are constants specific to each spectral window. The $a_{\mathrm{noise}}$ and $b_{\mathrm{noise}}$ values (B. Sierk, private communication) are provided in Table 1 as well.

**Table 1.** Setup of the CO2M spectrometer

| Band ID | spectral range [nm] | spectral resolution [nm] | spectral sampling ratio | $a_{\mathrm{noise}}$ [photons$^{-1}$cm$^2$ s nm sr] | $b_{\mathrm{noise}}$ |
|---------|--------------------|--------------------------|-------------------------|--------------------------------------|-----------|
| NIR     | 747-773   | 0.12 | 3 | $2.0 \times 10^{-8}$  | 19600  |
| SWIR1   | 1590-1675 | 0.30 | 3 | $1.32 \times 10^{-7}$ | 202500 |
| SWIR2   | 1990-2095 | 0.35 | 3 | $1.54 \times 10^{-7}$ | 202500 |

In our study, we consider two MAP instrument concepts, i.e. MAP-mod and MAP-band. Here, the MAP-mod instrument inherits from the SPEXone instrument (Hasekamp et al., 2019), which will fly on the NASA PACE mission, scheduled to launch in 2022. SPEXone will provide measurements in the visible between 385-770 nm at 5 viewing angles. In the MAP-mod concept, the polarimetric spectral resolution, which derives from the modulation period, becomes coarser at longer wavelengths while the radiance measurements can be obtained at the intrinsic spectral resolution of the instrument (Rietjens et al., 2015). Unlike a MAP-mod instrument that measures a continuous spectrum, a MAP-band instrument measures radiance and polarization at discrete spectral bands. Here, the spectral bands are specified close to the 3MI VNIR channels where both radiance and polarization are measured (Fougnie et al., 2018).

## 3 Methods

### 3.1 Joint MAP and spectrometer retrieval

To enable $XCO_2$ retrievals using MAP and spectrometer measurements in a synergistic way, we developed a joint retrieval algorithm. It is built upon an existing aerosol retrieval algorithm (Hasekamp et al., 2011; Fu and Hasekamp, 2018; Fu et al., 2020) to include features related to trace gas retrieval, with some spectrometer-specific functionalities incorporated in it. The joint retrieval tool can be used with either the MAP-mod or the MAP-band design. ~~Although t~~The algorithm is capable to simultaneously retrieve aerosol properties and the trace gas total columns ~~, in this work we are interested primarily in XCO₂ and do not discuss the retrieved aerosol properties~~.

In this retrieval, the concept of inverse modeling applies, in which state vector $\mathbf{x}$ is updated until it produces modeled measurements that fit the measurement vector $\mathbf{y}$ well enough. The modeled measurements or forward model $\mathbf{F}$ relates the state and measurement vectors via

$$\mathbf{y} = \mathbf{F}(\mathbf{x}, \mathbf{b}) + \epsilon_{\mathbf{y}} + \epsilon_{\mathbf{F}}, \tag{1}$$

where the terms $\epsilon_{\mathbf{y}}$ and $\epsilon_{\mathbf{F}}$ represent the measurement error and forward model error, while $\mathbf{b}$ constitutes auxiliary parameters needed to compute the forward model but are not retrieved. The $\mathbf{y}$ vector consists of the MAP and spectrometer measurements.

#### 3.1.1 Forward model

The forward model computes the Stokes vector, which describes the radiance and polarization state of light, at a certain wavelength and at a certain viewing angle for a specific atmospheric and geophysical scene. Degree of linear polarization (DLP) is then derived from the first three components of the vector $I, Q, U$, i.e. $\mathrm{DLP} = \sqrt{Q^2 + U^2}/I$, where $I$ constitutes the radiance. We consider both aerosols and trace gases in the model atmosphere. First, optical properties of molecules and aerosols are calculated. Aerosol optical properties are derived from the microphysical properties using tabulated kernels for a mixture of spheroids and spheres (Dubovik et al., 2006). The complex refractive index of aerosols is computed as a function of MAP wavelengths, but is assumed constant with wavelength inside a spectrometer window. Optical thickness due to molecular scattering is determined from the Rayleigh scattering cross section (Bucholtz, 1995). In the three spectrometer windows, molecular absorption features correspond to $O_2$ in the NIR band, $H_2O, CO_2$ in the two SWIR bands, and $CH_4$ in the SWIR1 band. Molecular absorption optical thickness is computed from the absorption cross-section values, which are pre-calculated from the latest spectroscopic databases (Tran et al., 2006; Rothman et al., 2009; Scheepmaker et al., 2013) and stored in a look-up table as a function of pressure, temperature and wavenumber.

Stokes parameters are computed from the optical properties via the radiative transfer model based on the work of Landgraf et al. (2001); Hasekamp and Landgraf (2002); Hasekamp and Landgraf (2005). For the polarimeter, Stokes parameters are calculated at each MAP wavelength. These modeled radiances and DLP directly represent the simulated MAP observations. To create spectrometer synthetic observations, optical thickness due to molecular and aerosol scattering and absorption and the radiance spectra are computed on a finely-sampled wavelength grid within each spectrometer window. Multiple scattering

contribution to the radiances is approximated by the linear-k approach (Hasekamp and Butz, 2008) to reduce computational time. The ~~measured~~ radiances are simulated by convolving the modeled radiances on the fine spectral grid with the instrument spectral response function, with random noise added afterwards. The instrument response function is modeled as a Gaussian with a Full Width at Half Maximum set to the spectral resolution listed in Table 1 and the noise follows the SNR defined in section 2.

The model atmosphere consists of 15 predefined height layers. Atmospheric vertical profiles of temperature, $H_2O, CO_2$, and $CH_4$ are provided as input. Aerosols consist of two modes, referred to as the fine and the coarse mode. The size distribution of each mode is quantified by a lognormal distribution (see Appendix A). To describe the vertical distribution of aerosols in the atmosphere, we adopt a Gaussian shape such that the number density of each aerosol mode at layer $k$ is given by

$$N_{0,k} = N_{aer} h(z_k) \Delta z_k, \tag{2}$$

where

$$h(z_k) = A \exp \left( -\frac{4(z_k - z_{aer})^2 \ln 2}{w_{aer}^2} \right). \tag{3}$$

$N_{aer}$ is the vertically integrated column number density, $A$ is a normalization constant, $z_k$ is the height of layer $k$, $w_{aer}$ is the width of the aerosol height distribution and $z_{aer}$ is the aerosol mean height (Butz et al., 2009, 2010).

The refractive index of each aerosol mode is defined by a linear combination of two aerosol types, such that the complex refractive index of a mode as a function of wavelength becomes

$$m(\lambda) = \sum_{l=1}^{2} c_l m_l(\lambda) \tag{4}$$

with $0.0 \leq c_l \leq 1.0$. $m_l(\lambda)$ is a wavelength-dependent complex refractive index for a certain aerosol type $l$, i.e. dust, inorganic matter (inorg), or black carbon (BC). The model size distribution and composition per mode do not vary with height. The aerosol particles are assumed to be a mixture of spheroids and spheres, where the proportion of the latter is characterized by the fraction of spheres ($f_{sphere}$).

To account for the reflection and polarization properties of the surface, the retrieval algorithm employs semi empirical bidirectional reflectance distribution function (BRDF) and bidirectional polarization distribution function (BPDF) models. The BRDF is characterized by the surface total reflectances $R_I$ that are modeled using a linear combination of kernels in the form

$$R_I(\lambda, \theta_\nu, \theta_0, \Delta\phi) = k_\lambda [1 + k_{geo} f_{geo}(\theta_\nu, \theta_0, \Delta\phi) + k_{vol} f_{vol}(\theta_\nu, \theta_0, \Delta\phi)] \tag{5}$$

(Litvinov et al., 2011), where $\theta_\nu, \theta_0$ are the viewing and solar zenith angles, and $\Delta\phi$ is the relative azimuth angle. We use the Ross-thick kernel for volumetric scattering kernel $f_{vol}$, and the Li-sparse kernel as geometric-optical scattering kernel $f_{geo}$ (Wanner et al., 1995). The BPDF is modeled according to the linear one-parameter model proposed by Maignan et al. (2009), with $\alpha$ being the only free parameter.

### 3.1.2 State vector

Although our main focus in this retrieval is XCO$_2$ , we also retrieve aerosol properties, together with surface attributes, and the total columns of CH$_4$ and H$_2$O. We take the input vertical profiles of the trace gases as a given and retrieve the total columns via scaling factors. Here, the prior and first guess of the scaling factor for each gas species are always 1.0, corresponding to the input total column. *Regarding the surface attributes, $k_\lambda$ at every measured MAP wavelength and for every spectrometer window, $k_{\mathrm{vol}}, k_{\mathrm{geo}}$, and $\alpha$ are considered the unknowns and are therefore state variables.* The majority of aerosol properties are included in the state vector; there are only 4 aerosol parameters that are not retrieved, i.e. $f_{\mathrm{sphere}}$, $z_{\mathrm{aer}}$ of the fine-mode aerosol, and $w_{\mathrm{aer}}$ for both modes. In our retrievals with synthetic data here, the four parameters are fixed to the true values. ~~Regarding the surface attributes, $k_\lambda$ at every measured MAP wavelength and for every spectrometer window, $k_{\mathrm{vol}}, k_{\mathrm{geo}}$, and $\alpha$ are considered the unknowns and therefore are part of the state vector.~~ *$f_{\mathrm{sphere}}$ of the fine mode is not fitted because non-spherical particles mostly relate to mineral dust which is predominantly in the coarse mode. The choice not to fit $z_{\mathrm{aer}}$ of the fine mode is appropriate for a situation with industrial aerosol (fine mode) in the boundary layer and an elevated dust layer (coarse mode). For other scenarios, this choice may not be optimal. For instance, it may be better to fit one value for $z_{\mathrm{aer}}$ that corresponds to both fine and coarse modes (Wu et al., 2016) in the case of biomass burning plumes. An investigation into how different choices of aerosol state variables could affect the retrieved XCO$_2$ errors is outside the scope of this paper and a subject of further study.* The complete list of the state variables, along with the prior values and prior uncertainties, is given in Table 2.

### 3.1.3 Inversion procedure

The goal of the retrieval is to find $\mathbf{x}$ which would result in $\mathbf{F}(\mathbf{x},\mathbf{b})$ that best matches $\mathbf{y}$. This is achieved by minimizing the cost function

$$\hat{\mathbf{x}} = \underset{\mathbf{x}}{\arg\min}\,(\|\mathbf{S_y}^{-\frac{1}{2}}(\mathbf{y}-\mathbf{F}(\mathbf{x},\mathbf{b}))\|^2 + \gamma^2\|\mathbf{W}(\mathbf{x}-\mathbf{x_a})\|^2) \tag{6}$$

$\mathbf{x_a}$ is the prior state vector, $\mathbf{W}$ is a weight matrix, while $\gamma$ is the Phillips-Tikhonov regularization parameter (Phillips, 1962; Tikhonov, 1963). Regularization is needed to obtain a stable solution since the inverse problem is ill-posed. The weight matrix $\mathbf{W}$ is constructed in such a way that it brings all state vector elements to the same order of magnitude; it is also used in the inversion to give more freedom to some state vector parameters, for which the prior information is assumed less reliable compared to the others. Here, $\mathbf{W} = \mathbf{S_a}^{-\frac{1}{2}}$ such that for $\gamma = 1$, Eq. (6) reduces to the Optimal Estimation cost function.

Due to the non-linearity of the forward model, the minimization problem is solved in an iterative manner. At every iteration step, $\mathbf{F}(\mathbf{x})$ is linearized such that at iteration $n$:

$$\mathbf{F}(\mathbf{x}_{n+1},\mathbf{b}) \approx \mathbf{F}(\mathbf{x}_n,\mathbf{b}) + \mathbf{K}(\mathbf{x}_{n+1}-\mathbf{x}_n). \tag{7}$$

$\mathbf{K}$ is the Jacobian matrix containing partial derivatives of forward model element $F_i$ with respect to state variable $x_j$, i.e.

$$K_{ij} = \frac{\partial F_i(\mathbf{x}_n)}{\partial x_j}. \tag{8}$$

**Table 2.** State variables in the joint retrieval

|  | State parameter | Prior | Prior error |
|---|---|---|---|
| trace gas | $CO_2$ scaling factor | 1.0 | 1.0 |
|  | $H_2O$ scaling factor | 1.0 | 1.0 |
|  | $CH_4$ scaling factor | 1.0 | 1.0 |
| fine-mode aerosol | $r_{\mathrm{eff}}$ [$\mu$m] | 0.2 | 0.1 |
|  | $v_{\mathrm{eff}}$ | 0.2 | 0.05 |
|  | $c_1$ (inorg) | 0.9 | 0.1 |
|  | $c_2$ (BC) | 0.1 | 0.1 |
|  | $\tau$ at 550 nm | 0.2 | 1.0 |
| coarse-mode aerosol | $r_{\mathrm{eff}}$ [$\mu$m] | 1.5 | 1.0 |
|  | $v_{\mathrm{eff}}$ | 0.6 | 0.1 |
|  | $c_1$ (dust) | 0.5 | 0.1 |
|  | $c_2$ (inorg) | 0.5 | 0.1 |
|  | $\tau$ at 550 nm | 0.05 | 0.2 |
|  | $f_{\mathrm{sphere}}$ | 0.05 | 0.5 |
|  | $z_{\mathrm{aer}}$ [m] | 6500 | 4000 |
| surface properties | $k_{\mathrm{geo}}$ | 0.0 | 0.25 |
|  | $k_{\mathrm{vol}}$ | 0.0 | 1.0 |
|  | multiple $k_\lambda$ | 0.0 | 0.5 |
|  | $\alpha$ | 1.0 | 2.0 |

The linearization in Eq. (7) modifies the optimization problem to

$$\tilde{\mathbf{x}}_{n+1} = \underset{\mathbf{x}}{\arg\min} \left( \|\tilde{\mathbf{y}} + \tilde{\mathbf{K}}\tilde{\mathbf{x}}_n - \tilde{\mathbf{K}}\tilde{\mathbf{x}}\|^2 + \gamma^2 \|\tilde{\mathbf{x}} - \tilde{\mathbf{x}}_{\mathbf{a}}\|^2 \right), \tag{9}$$

with

$$\tilde{\mathbf{x}} = \mathbf{W}\mathbf{x}, \tag{10}$$

$$\tilde{\mathbf{y}} = \mathbf{S}_{\mathbf{y}}^{-\frac{1}{2}} (\mathbf{y} - \mathbf{F}(\mathbf{x}_n)), \tag{11}$$

$$\tilde{\mathbf{K}} = \mathbf{S}_{\mathbf{y}}^{-\frac{1}{2}} \mathbf{K}\mathbf{W}^{-1}. \tag{12}$$

The solution is found using an iterative Gauss-Newton method and expressed in terms of the departure $\Delta\tilde{\mathbf{x}}$ from $\tilde{\mathbf{x}}_n$ (Rodgers, 2000). Given that the forward model is highly non linear, the retrieval could diverge when the current $\tilde{\mathbf{x}}_n$ is far from the solution. To avoid that, we reduce the step size $\Delta\tilde{\mathbf{x}}$ by applying the $\Lambda$ factor, i.e.

$$\tilde{\mathbf{x}}_{n+1} = \tilde{\mathbf{x}}_n + \Lambda(\Delta\tilde{\mathbf{x}}), \tag{13}$$

with

$$\Delta\tilde{\mathbf{x}} = (\tilde{\mathbf{K}}^T\tilde{\mathbf{K}} + \gamma^{\mathbf{2}}\mathbf{I})^{-1}[\tilde{\mathbf{K}}^T\tilde{\mathbf{y}} - \gamma^2(\tilde{\mathbf{x}}_n - \tilde{\mathbf{x}}_{\mathbf{a}})] \tag{14}$$

At each iteration step, we compute a fast and simplified forward model using a combination of 5 possible $\gamma$ and 10 possible $\Lambda$ values. $\gamma$ is varied from 0.1 to 5 whereas $\Lambda$ is varied from 0.1 and 1.0. The combination of $\gamma$ and $\Lambda$ that delivers the best match to the measurements, via $\chi^2$ assessment, is adopted in Eq. (14) to compute the step size. The iteration in the inversion starts with a first guess $\tilde{\mathbf{x}}_1$ that is generated via a look-up table retrieval.

## 3.2 Linear error analysis

Linear error analysis allows $XCO_2$ errors to be derived in a way that mimics as close as possible an iterative retrieval method and so is particularly useful in performing the MAP requirement study. Linear error analysis delivers an aerosol-induced $XCO_2$ error $\langle\Delta XCO_2\rangle$ that is representative in the statistical sense. It is impractical to derive this error using iterative retrieval method, given the large number of scenarios and instrument setups that we evaluate in the requirement study. The basic principles of the error analysis that we employ can be found in Rodgers (2000). Here we describe the mathematical formalism of the analysis.

In order to estimate $\langle\Delta XCO_2\rangle$ , the error analysis follows a two-step approach. *Splitting the analysis in two steps helps to illustrate how the uncertainties in the aerosol properties contribute to $XCO_2$ errors.* The first step (step 1) corresponds to the aerosol retrieval using MAP. Here the uncertainties on aerosol parameters are derived. The second step (step 2) represents $XCO_2$ retrieval using spectrometer measurements where the aerosol uncertainties from step 1 are propagated to result in $\langle\Delta XCO_2\rangle$ . In both steps, aerosols are parametrized in the same way as in the section 3.1.1. As in section 3.1.2, we do not retrieve $w_{\mathrm{aer}}$ of both modes, $z_{\mathrm{aer}}$ and $f_{\mathrm{sphere}}$ of the fine mode, which leaves us with 12 aerosol parameters in the state vector.

We compute for each scenario two Jacobian matrices. One of the Jacobians is associated to retrieval step 1 with a given MAP setup ($\mathbf{K}_{\mathrm{MAP}}$), and the other belongs to retrieval step 2 ($\mathbf{K}_{\mathrm{spc}}$). State variables in step 1 comprise aerosol and surface parameters (see section 3.1.1 for details). Measurement variables of step 1 consist of radiances and DLP; their composition depends on the MAP instrument setup being used. State vector of retrieval step 2 follows that used in the spectrometer-only retrieval described in section 3.3, with the exception of the aerosol parameters, i.e. here we use the bimodal lognormal model (Eq. A1) and fit 12 parameters (Table 2). Both $\mathbf{K}_{\mathrm{spc}}$ and $\mathbf{K}_{\mathrm{MAP}}$ are calculated at the true state vector values and each of them contains derivatives of the corresponding measurements with respect to 12 aerosol parameters.

Errors on the retrieved aerosol properties from step 1 comprise the smoothing errors and the MAP-measurement-noise-induced error (retrieval noise). The smoothing error is formulated as

$$\mathbf{S}_{\mathrm{MAP}}^{\mathrm{sm}} = (\mathbf{G}_{\mathrm{MAP}}\mathbf{K}_{\mathrm{MAP}} - \mathbf{I})\mathbf{S}_{\mathrm{a,MAP}}(\mathbf{G}_{\mathrm{MAP}}\mathbf{K}_{\mathrm{MAP}} - \mathbf{I})^T, \tag{15}$$

whereas the retrieval noise reads

$$\mathbf{S}_{\mathrm{MAP}}^{\mathrm{ns}} = \mathbf{G}_{\mathrm{MAP}}\mathbf{S}_{\mathrm{y,MAP}}\mathbf{G}_{\mathrm{MAP}}^T. \tag{16}$$

$S_{a,MAP}$ is a MAP prior error covariance matrix, which is a diagonal matrix containing squared prior errors. The values of prior errors that we use here for the aerosol and surface variables are the same as those stated in Table 2. $S_{y,MAP}$ is a measurement error covariance matrix containing squared values of MAP measurement errors along the diagonal axis. The individual radiometric and polarimetric measurements are assumed uncorrelated, so the off-diagonal elements are set to zero. $G_{MAP}$ is the gain matrix that relates the MAP measurement errors with the noise in MAP state parameters and it follows that

$$G_{MAP} = (K_{MAP}^T S_{y,MAP}^{-1} K_{MAP} + S_{a,MAP}^{-1})^{-1} K_{MAP}^T S_{y,MAP}^{-1}. \tag{17}$$

The total aerosol (posterior) error covariance matrix is then

$$S_{MAP}^{tot} = S_{MAP}^{ns} + S_{MAP}^{sm}. \tag{18}$$

Cropping $S_{MAP}^{tot}$ to keep only variances and covariances of the 12 aerosol parameters (i.e. excluding the surface parameters) results in $S_{aer}^{tot} \in \mathbb{R}^{12 \times 12}$.

The matrix $S_{aer}^{tot}$, which represents the total uncertainties in aerosol parameters retrieved from MAP measurements, is then passed on to the second part of the error analysis. At this stage, the aerosol errors are mapped into spectrometer measurement errors using $K_{spc}$. The spectrometer measurement errors are in turn propagated into the errors of the step-2 state variables using the spectrometer gain matrix $G_{spc}$. The mathematical expression for this propagation chain in step 2 is

$$S_{spc}^{aer} = G_{spc} K_{spc}^{aer} S_{aer}^{tot} (K_{spc}^{aer})^T G_{spc}^T, \tag{19}$$

where

$$G_{spc} = (K_{spc}^T S_{y,spc}^{-1} K_{spc} + \gamma^2 W)^{-1} K_{spc}^T S_{y,spc}^{-1}. \tag{20}$$

$K_{spc}^{aer}$ is a subset of $K_{spc}$ containing derivatives with respect to aerosol properties only. The regularization term $\gamma^2 W$ is adjusted accordingly to arrive at the typical degrees of freedom between 2 and 3 for aerosol parameters (Guerlet et al., 2013; Wu et al., 2019). $S_{y,spc}^{-1}$ is a diagonal matrix containing the squared values of the spectrometer measurement noise.

Finally, $\langle \Delta XCO_2 \rangle$ is obtained from

$$\langle \Delta XCO_2 \rangle = \sqrt{S_{spc}^{aer}[1,1]}, \tag{21}$$

where the first diagonal element $S_{spc}^{aer}[1,1]$ is the variance of the $CO_2$ total column. Note that $\langle \Delta XCO_2 \rangle$ includes the error contribution from the MAP noise, but not from the spectrometer noise.

## 3.3 Spectrometer-only retrieval

To perform $XCO_2$ retrievals using spectrometer measurements only, we employ the RemoTeC algorithm which was designed for greenhouse gas retrievals using satellite observations (Butz et al., 2009, 2010). The forward modelling adopts the latest version developed for TROPOMI (Hu et al., 2016) and the inversion procedure largely follows Butz et al. (2012). In what follows, we highlight aspects that are specific to this work.

As in the joint retrieval, we retrieve the total column of $CO_2$, along with the total columns of two other trace gases i.e. $H_2O$ and $CH_4$, via scaling factors. Prior values of the total columns are derived from the input profiles, i.e. corresponding to scaling factors of 1. The input profiles of temperature, pressure and the gases are the same as those used in the joint retrieval.

With only spectrometer data available, we resort to a simplifed approximation of aerosols in the retrieval. In the forward model, aerosols are described by a simple model where the size distribution is parametrized by a monomodal power-law function. The power-law distribution is prescribed in Mishchenko et al. (1999) and it reads:

$$n(r) = \begin{cases} B, & \text{if } r < r_1. \\ B(r/r_1)^{-\beta}, & \text{if } r_1 < r \leq r_2. \\ 0, & \text{if } > r_2. \end{cases} \tag{22}$$

$B$ is the normalization constant and $r$ is the aerosol particle radius. $r_1$ is fixed to 0.1 $\mu$m and $r_2$ to 10 $\mu$m. The height distribution

follows Equations (2) and (3) with $w_{aer}$ set to 2 km. The real and imaginary refractive indices are kept constant at 1.4 and $-0.003$, respectively, in all three spectral windows. Aerosol properties that are retrieved include the optical depth $\tau$ at 765 nm (prior=0.1), the size distribution parameter $\beta$ (prior=4.0), and the mean height $z_{aer}$ (prior=3000 m). This simplified aerosol model is adopted by e.g. Butz et al. (2009); Butz et al. (2012); Hu et al. (2016); Wu et al. (2019, 2020) as the standard approach to account for aerosol effects on the retrieved $XCO_2$ and $XCH_4$.

The reflection at the Earth surface is assumed to be Lambertian. Surface reflectance is included in the state vector via the albedo and its wavelength dependence in each window, which are modelled as a first-order polynomial. The prior for the albedo is the Lambertian-equivalent albedo corresponding to the maximum radiance measured in the retrieval window in question. The slope of the polynomial (wavelength dependence of the albedo) is given a prior of 0.0. Additionally, for each spectral window a spectral shift parameter is retrieved with a prior of 0.0. In total, the state vector consists of 15 variables, i.e. the total

columns of three trace gases $(CO_2, CH_4, H_2O)$, 3 aerosol properties, and for each spectral window 2 albedo parameters and 1 spectral shift parameter.

## 4   Ensemble of synthetic scenes

We construct an ensemble of 500 synthetic scenes, characterized by different combinations of trace gas and aerosol content, surface albedo, and solar zenith angle (SZA). Every scene is generated by randomly varying those atmospheric and geophysical

properties. The random value is drawn from a uniform distribution within a specific interval. *Vertical profiles of pressure, temperature, water vapor, and trace gases are adopted from the AFGL atmospheric profiles (Shettle and Fenn, 1979), with $CO_2$ scaled up such that the total column is 400 ppm.*

Given the spectral windows of the CO2M spectrometer, the radiance spectra include absorption features due to $CO_2$, $H_2O$, and $CH_4$. In this ensemble, the vertical profile of the individual gas is fixed and we vary the total column, which is represented

by the scaling factor. The scaling factors of $CO_2$, $H_2O$, and $CH_4$ are varied by 5%, 3%, and 6%, corresponding to the intervals [0.95,1.05], [0.97,1.03], and [0.94,1.06], respectively. A scaling factor of 1.0 means that the total column is obtained from

vertically integrating the column number density given in the atmospheric input profile, which for $CO_2$ amounts to 400 ppm. Given that the scaling factors in the scenes are randomized, the true total columns are by intention different from the prior values in our retrieval exercises.

Aerosols in every scene are constructed to consist of the fine and the coarse mode. The size distribution of each mode is quantified by a lognormal distribution (Eq. A1). The vertical distribution of an aerosol mode in the atmosphere follows a Gaussian shape (Eq. 3). The refractive index of each aerosol mode is defined by Eq. (4) where the coarse mode is composed of the dust and inorganic types, while the fine mode is made up of inorganic matter and black carbon. The fine mode is set up to consist entirely of spheres, i.e. fraction of spheres $f_{sphere} = 1.0$, for which the Mie theory applies. The coarse-mode

particles are described by a mixture of spheroids and spheres, following Dubovik et al. (2006). Aerosol size distribution and composition of each mode are constant with height. Most of the aerosol parameters are varied randomly, while a few are held fixed. Table 3 provides the complete list of the aerosol properties and the corresponding intervals from which random values are drawn, or the corresponding values for the fixed parameters.

Table 3. Aerosol properties in the ensemble. The numbers in square brackets specify the interval from which a random value is drawn, whereas a single number indicates a fixed value.

| Aerosol parameters | fine mode | coarse mode |
|---|---|---|
| $r_{eff}$ [$\mu$m] | [0.1,0.3] | [0.65,3.4] |
| $v_{eff}$ | [0.1,0.3] | [0.45,0.65] |
| $c_1$ | [0.887,0.975] (inorg) | [0.439,0.512] (dust) |
| $c_2$ | [0.0,0.05] (BC) | [0.439,0.512] (inorg) |
| $\tau$ at 765 nm | [0.002,0.52] | [0.0048,0.32] |
| $f_{sphere}$ | 1.0 | [0.0,0.5] |
| $z_{aer}$ [m] | 1000 | [1000,8500] |
| $w_{aer}$ [m] | 2000 | 2000 |

Solar zenith angle is allowed to take any value between 10 and 70 degrees. Surface albedo $\rho$ is determined from the combi-
nation of albedos of two surface types, i.e. soil and vegetation. More specifically,

$$\rho = f\rho_{soil} + (1-f)\rho_{veg}, \tag{23}$$

in which $\rho_{soil}$ and $\rho_{veg}$ are provided in a tabular form as a function of wavelength, and $f$ is varied between 0 and 1.

The simulated spectra for the spectrometer-only retrieval (sections 3.3 and 5) are not identical to the simulated spectrometer measurements for the joint retrieval (sections 3.1 and 7). This is because the surface descriptions used in the two retrieval
approaches differ. To isolate aerosol-induced errors, we use the same surface model in the forward simulation as in the retrieval. Consequently, we use the Lambertian surface assumption to generate spectra for the spectrometer-only retrieval, where $\rho$ at 765, 1600, and 2000 nm are assigned as the albedo in NIR, SWIR1 and SWIR2 windows, respectively. Albedo within a spectral window is not wavelength dependent. For the joint retrieval exercise, synthetic spectra of both the polarimeter and the

spectrometer are generated by employing the BRDF and BPDF models (Eq. 5, section 3.1.1), with $k_{\mathrm{vol}} = 1.0$, $k_{\mathrm{geo}} = 0.1$, and
$\alpha = 1.0$ (non-Lambertian surface). The values of $k_\lambda$ are set to $\rho$ at 765, 1600, and 2000 nm for the three spectral windows ($k_\lambda$
is fixed within a spectrometer window) and to $\rho$ at each MAP wavelength.

Finally, we add random realizations of the instrument noise to the ~~simulated~~ *synthetic* measurements. It is this noisy spectra
that are given as input data for the retrievals. For the spectrometer, the noise follows the formulation in section 2. For MAP,
the noise on radiance is typically a few percent, while the noise on DLP is of the order of $10^{-3}$ (the determination of the
appropriate noise level is a part of the requirement study in section 6).

## 5   XCO$_2$ retrieval using CO2M spectrometer measurements only

Here we present the results of RemoTeC iterative retrievals (section 3.3) of XCO$_2$ on 500 synthetic scenes described in sec-
tion 4, using only spectrometer observations (section 2). The goodness-of-fit of the retrieval results is evaluated via $\chi^2 =$
$\frac{1}{N_{\mathrm{meas}}} \sum_{i=1}^{N_{\mathrm{meas}}} \left( \frac{y_i - F_i}{e_i} \right)^2$. $N_{\mathrm{meas}}$ is the total number of measurements, $y_i$, $F_i$, $e_i$ represent the (synthetic) measurements, the
forward model, and the measurement uncertainties, respectively. Since we use noisy synthetic measurements, we expect that
a successful and converged retrieval would have a $\chi^2$ of around 1. Here, we impose a $\chi^2$ criterion by applying a threshold of
1.5. Only for converged retrievals with $\chi^2 \leq 1.5$ do we assess the retrieval error $\Delta$XCO$_2$ and how it depends on some aerosol,
surface properties and SZA. $\Delta$XCO$_2$ is the residual XCO$_2$ , i.e the retrieved minus the true total column, so it represents the
combined effect of aerosol-induced and spectrometer-noise-induced errors on the retrieved XCO$_2$ .

Out of 500 retrievals, 343 converge and meet the $\chi^2$ criterion (convergence rate of 69%). Figure 1 plots $\Delta$XCO$_2$ of the
343 retrievals as a function of true aerosol optical depth, aerosol height, SZA, and albedo. *As a side remark, scatter plots
of $\Delta$XCO$_2$ against the retrieved $\tau$ , aerosol height, and albedo show similar trends; here we focus on discussing Figure 1.*
$\Delta$XCO$_2$ is evidently sensitive to the changes in aerosol optical depth, aerosol height and the SZA. In what follows, we use the
median value of $\Delta$XCO$_2$ to describe the bias. For $\tau < 0.07$, $\Delta$XCO$_2$ are relatively unbiased and confined to within $\pm 1.5$ppm.
Starting $\tau \sim 0.1$, the scatter in $\Delta$XCO$_2$ becomes much larger with an overall bias of $\sim 1.9$ ppm at $\tau \sim 0.5$. Beyond $\tau$ of 0.6,
there are notably fewer converged retrievals; almost all of them have positive $\Delta$XCO$_2$ . Like the optical depth, aerosol height
is fitted in the retrieval. For scenes with a true coarse-mode aerosol height up to 1700 m, $\Delta$XCO$_2$ is found between -2.7 and
+2.6 ppm with a relatively small positive bias of 0.3 ppm. However, as the true coarse-mode aerosol height increases, so do
the scatter and the bias in XCO$_2$ . At heights > 7000 m, $\Delta$XCO$_2$ is distributed between -5.9 and +7.5 ppm with a bias of 1.1
ppm. An opposite trend is seen for SZA where the highest bias is observed for the lowest SZA between 10°and 30°. There, the
median value of $\Delta$XCO$_2$ is 1.7 ppm. Between SZA of 60°and 70°, the retrieval bias is the smallest at 0.6 ppm. $\Delta$XCO$_2$ scatter
shows a slight reduction with increasing SZA. Dependency of $\Delta$XCO$_2$ on the surface albedo at 2000 nm is not apparent in our
ensemble; there is a an overall positive bias without any noteworthy correlation. The same can be said about the behaviour of
$\Delta$XCO$_2$ with respect to albedo in the other two spectral windows and about the behaviour of $\Delta$XCO$_2$ as a function of blended
albedo (following the definition in Wunch et al. (2011)).

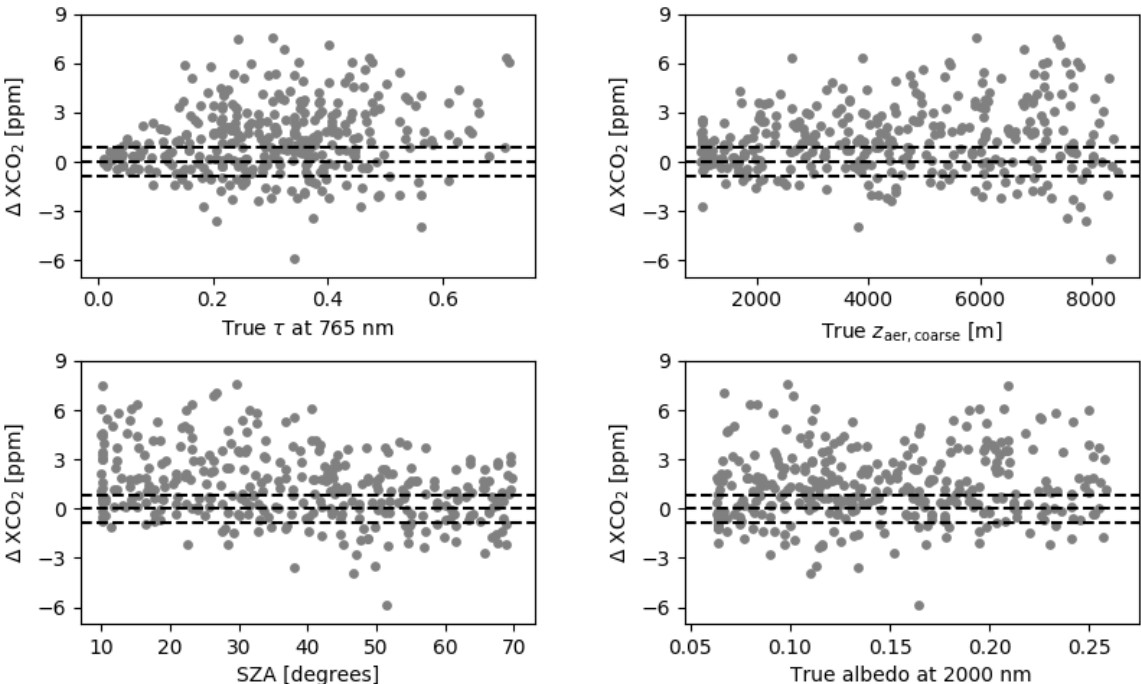

**Figure 1.** Residual XCO$_2$ from the converged retrievals using spectrometer measurements only, shown as a function of total aerosol optical depth $\tau$, coarse-mode aerosol height, SZA, and albedo. The input spectra are generated according to the ensemble of synthetic scenes (Section 4). The three dashed horizontal lines indicate $\Delta$XCO$_2$ at -0.86, 0, and 0.86 ppm.

The trends of $\Delta$XCO$_2$ that we see here are largely consistent with those in Butz et al. (2009). In their 3-band noise-free retrieval exercise, they observed a positive bias in the residual XCO$_2$ error distribution, that the scatter in XCO$_2$ errors increased with aerosol optical thickness, and that the XCO$_2$ errors did not show a pronounced dependence on surface albedo. For most of the cases with $\tau \leq 0.5$, their XCO$_2$ errors were confined to less than 1%. Our results reflect all of those findings. Butz et al.
evaluated their ensemble at two SZA values, i.e. 30° and 60°, and they found that a positive bias in samples with SZA=60° was stronger than for samples with SZA=30°. This may seem to contradict our result that shows a diminishing positive bias with increasing SZA, but one should note the use of different statistical samplings in our and their analyses, which prevents a direct comparison.

To minimize outlier effects on the statistics, we choose to evaluate the bias and the spread of the $\Delta$XCO$_2$ distribution using
percentiles. We adopt the median (the 50th percentile) as a measure of the bias and $P_{SD} = 0.5 \times (P(84.1) - P(15.9))$ as a measure of the spread, where P(15.9) and P(84.1) are the 15.9th and the 84.1th percentiles. For a normal distribution, $P_{SD}$ reduces to the standard deviation, within which $\sim$68% of the instances fall. From the 343 retrievals, the median $\Delta$XCO$_2$ is found at 1.12 ppm and $P_{SD}$ is equal to 2.07 ppm. Our median value is in between the 3-band retrievals of Butz et al. (2009) for SZA=30° (0.42 ppm) and SZA=60° (1.2 ppm), while our $P_{SD}$ is above the standard deviation found by Butz et al. (1.29
for SZA=30° and 1.42 ppm for SZA=60°). Our higher $P_{SD}$ could be attributed, at least partially, to the additional instrument

noise in the simulated measurements since Butz et al. used synthetic measurements without noise. *If we filter out converged retrievals with retrieved $\tau$ (at 765 nm) > 0.3, only 232 retrievals are retained with a bias and $P_{SD}$ of 1.13 ppm and 1.95 ppm. Lowering the $\tau$ threshold further to 0.2 means only 136 retrievals are selected with a slightly smaller bias and $P_{SD}$ (0.9 and 1.66 ppm).*

As mentioned above, the accuracy and precision requirements of the CO2M mission are 0.5 and 0.7 ppm, respectively. The quadratic sum amounts to a total $XCO_2$ uncertainty of 0.86 ppm. With a $P_{SD}$ above 2 ppm *(without $\tau$ filtering)*, $XCO_2$ retrievals based on only spectrometer measurements do not meet the mission requirements by a very wide margin ~~(note that we do not apply post-retrieval filtering here)~~. *For real data, it is certainly possible to decrease $XCO_2$ errors to within the mission requirements through heavy post-retrieval filtering and bias corrections, but this would mean a significant reduction in data*

*volume.* In section 7, we investigate the improvement in $XCO_2$ accuracy and precision when a MAP instrument is flown aboard the CO2M satellite.

## 6    MAP requirement analysis

Before we can assess the contribution of a MAP instrument in improving the $XCO_2$ retrieval performance, we first determine the required specifications for such an instrument given the precision and accuracy threshold of the CO2M mission.

As stated in section 2, two alternative MAP instrument concepts are being considered, i.e. MAP-mod and MAP-band. For each concept, we look into multiple instrument configurations to search for the optimal one that meets the CO2M mission requirement by performing linear error analyses (section 3.2). A linear error analysis delivers $\langle \Delta XCO_2 \rangle$ that represents aerosol-induced $XCO_2$ errors, which consist of both systematic and random components. The MAP instrument noise is included in $\langle \Delta XCO_2 \rangle$ but the spectrometer noise is not accounted for. For a total error budget of 0.86 ppm (quadratic sum of the CO2M

accuracy and precision requirements), here we assume equal error contributions from the MAP and from the spectrometer. This means we allocate 0.6 ppm for $\langle \Delta XCO_2 \rangle$ and leave 0.6 ppm for other error sources, e.g. spectrometer noise (the quadratic sum of 0.6 ppm and 0.6 ppm is 0.85 ppm).

We evaluate the performance of MAP instrument setups with respect to three aspects, i.e. radiance and polarization measurement uncertainties, number of viewing angles and the wavelength range. For this purpose, the linear error analysis is applied

to a generic set of study scenarios involving a variety of aerosol and surface properties. Below, we define the study cases, followed by the requirement analysis for the MAP-mod concept, which results in the MAP-mod baseline setup. Afterwards, we present the baseline setup for the MAP-band concept that we determine through a separate error analysis similar to that for the MAP-mod.

### 6.1    Study cases

We introduce three aerosol cases that form the basis of the scenarios used to derive the requirements. They are referred to as 'case 1', 'case 2', and 'case 3'. In all cases, the aerosols are modeled according to the bimodal lognormal size distribution, Gaussian height distribution, and the linear superposition of complex refractive index (Equations A1, 2-4), i.e. the same as the

parametrizations used to build the ensemble of synthetic scenes (section 4). The fine mode aerosol is always located at 1 km height in all cases. Case 1 is designed to mimic boundary layer aerosols where the fine and coarse mode aerosols coincide at 1 km. In case 2, the coarse mode represents an elevated layer at 8 km. Case 3 is a mid-troposphere case where the coarse mode is located at 4 km. All fine mode particles are spherical and non-absorbing, whereas the coarse mode constitute dust particles with $f_{sphere} = 0.05$. To account for the effects of aerosol load on $XCO_2$ retrieval, the aerosol optical depth $\tau$ of either the fine or the coarse mode is varied to 5 different values in each case. The summary of the aerosol properties for each case is given in Table 4.

**Table 4.** Aerosol properties adopted in the study cases

| Aerosol parameters | Case 1 | | Case 2 / 3 | |
|---|---|---|---|---|
| | fine mode | coarse mode | fine mode | coarse mode |
| $r_{eff}$ [$\mu$m] | 0.12 | 1.6 | 0.12 / 0.2 | 1.6 |
| $v_{eff}$ | 0.2 | 0.6 | 0.2 | 0.6 |
| $f_{sphere}$ | 1.0 | 0.05 | 1.0 | 0.05 |
| $c_1$ | 1.0(inorg) | 1.0(dust) | 1.0(inorg) | 1.0(dust) |
| $c_2$ | 0.0(BC) | 0.0(inorg) | 0.0(BC) | 0.0(inorg) |
| $z_{aer}$ [m] | 1000 | 1000 | 1000 | 8000 / 4000 |
| $w_{aer}$ [m] | 2000 | 2000 | 2000 | 2000 |
| $\tau$ at 765 nm | 0.05,0.1 0.15,0.25 0.5 | 0.02 | 0.2 | 0.02,0.04 0.06,0.10, 0.15 |

We consider two types of land surface, i.e. soil and vegetation. These are the basic surface types used to create the 500 synthetic scenes (section 4). Albedo values at 765 nm, 1600 nm, 2000 nm for the soil are 0.139,0.298,0.259, and for the vegetation type 0.450,0.230,0.063. We perform the error analysis for two solar zenith angles 30° and 60°. In total, the combination of 2 surface types, 2 SZA values and 15 aerosol variations results in 60 scenarios.

## 6.2 MAP-mod instrument

### 6.2.1 Radiometric and polarimetric uncertainties

To examine the sensitivity of $XCO_2$ estimates to MAP radiometric and polarimetric uncertainties, we vary $\mathbf{S}_{y,MAP}$ in Equations (16) and (17) and perform the error analysis. As a starting point, we adopt a setup that is similar to SPEXone (Hasekamp et al., 2019). For this exercise, we assume 5 viewing angles at 0, ±40, ±60 degrees. The spectral range extends from 385 to 765 nm with a fixed radiance spectral resolution of 5 nm, and a DLP spectral resolution of 15 nm at 395 nm and 30nm at 765 nm. This spectral arrangement corresponds to 77 radiance and 19 DLP measurements. With 5 viewing angles, the total number of measurements becomes 480. For each scenario, we compute $\langle \Delta XCO_2 \rangle$ that corresponds to each combination of 4

values of radiance uncertainties $\Delta I/I$ and 11 different values of DLP uncertainties $\Delta$DLP . $\Delta I/I$ ranges from 1% to 4% and $\Delta$DLP ranges from 0.001 to 0.05.

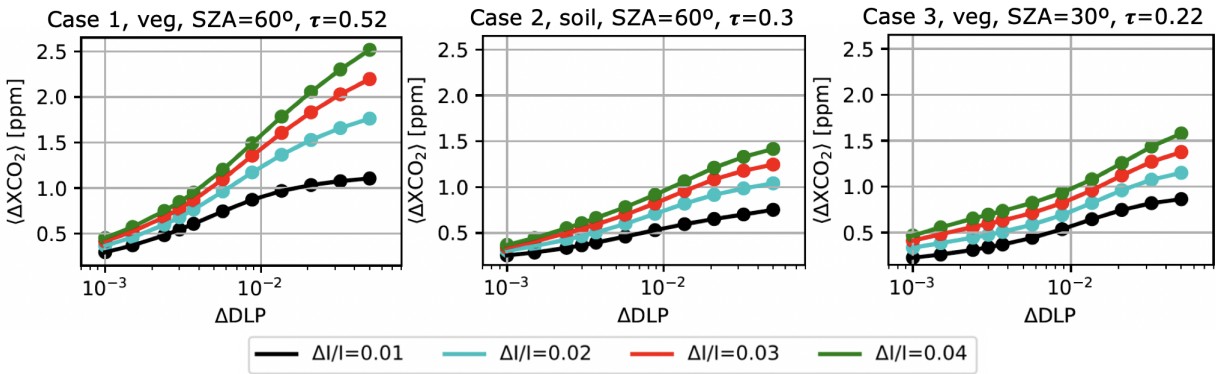

**Figure 2.** $\langle\Delta\text{XCO}_2\rangle$ as a function of $\Delta I/I$ and $\Delta$DLP , shown for scenarios where the errors are among the largest for each individual case 1,2, and 3. Here, $\tau$ indicates the sum of the fine- and the coarse-mode aerosol optical depth at 765 nm.

Figure 2 shows examples of $\langle\Delta\text{XCO}_2\rangle$ for all combinations of $\Delta I/I$ and $\Delta$DLP that we investigate. We present different

sets of surface type, SZA, and $\tau$ that deliver among the largest $\langle\Delta\text{XCO}_2\rangle$ in each aerosol case. The figure demonstrates that $\text{XCO}_2$ uncertainties increase, as expected, with increasing DLP and radiance errors, and that holds true in all other scenarios as well. $\langle\Delta\text{XCO}_2\rangle$ can be as high as ~ 2.4 *2.52* ppm for the highest $\Delta I/I$ and $\Delta$DLP considered here. It can be seen that the measurement errors that correspond to the allocated $\langle\Delta\text{XCO}_2\rangle$ of 0.6 ppm are a combination of $\Delta I/I$ of about 2-3% and $\Delta$DLP of around 0.003.

Figure 3 presents $\langle\Delta\text{XCO}_2\rangle$ for all study scenarios when $\Delta$DLP is fixed to 0.003, and compares the retrieval performance between $\Delta I/I$ =2% and $\Delta I/I$ =3%. Generally speaking, the difference in $\langle\Delta\text{XCO}_2\rangle$ for both $\Delta I/I$ values is relatively small. When radiance and DLP errors are not greater than 2% and 0.003, respectively, $\langle\Delta\text{XCO}_2\rangle$ does not get higher than 0.6 ppm except for 3 scenarios; in most cases, $\langle\Delta\text{XCO}_2\rangle$ is around or lower than 0.5 ppm. For $\Delta I/I$ of 3%, coupled with $\Delta$DLP =0.003, $\langle\Delta\text{XCO}_2\rangle$ increases beyond 0.6 ppm in a few scenarios by a relatively small margin. Among these scenarios, the highest

$\langle\Delta\text{XCO}_2\rangle$ is found for the case-1,vegetation,SZA= $60°$,$\tau$ =0.52 scenario where it is just under 0.8 ppm; for the other scenarios, $\langle\Delta\text{XCO}_2\rangle$ varies between 0.6-0.7 ppm. Given that the improvement in $\Delta I/I$ from 3% to 2% is a major technical challenge while the reduction in $\text{XCO}_2$ uncertainty is only marginal, we adopt the more relaxed requirement here ($\Delta I/I$ =3% and $\Delta$DLP =0.003).

### 6.2.2   Number of viewing angles

If we change the number of viewing zenith angles (VZAs), we effectively add or remove measurements and this would certainly influence the aerosol and hence the $\text{XCO}_2$ retrievals. A number of studies suggest that 5 viewing angles are sufficient for aerosol retrieval (Hasekamp and Landgraf, 2007; Wu et al., 2015; Xu et al., 2017; Hasekamp et al., 2019). Here, we vary the

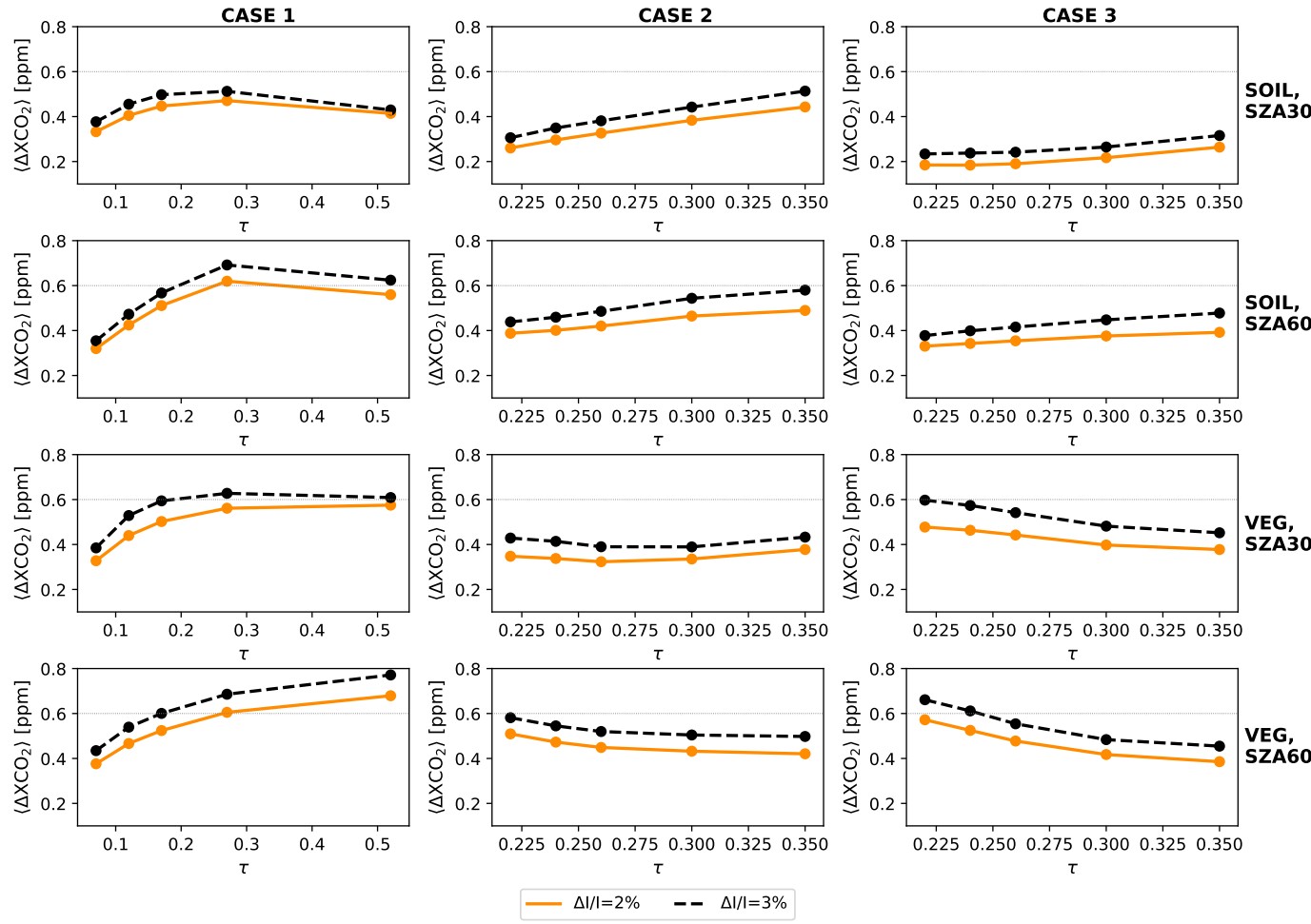

**Figure 3.** $\langle \Delta \mathrm{XCO}_2 \rangle$ as a function of total aerosol optical depth at 765 nm for all 60 study scenarios. The two lines in each panel show $\langle \Delta \mathrm{XCO}_2 \rangle$ for two different $\Delta I / I$ values, as computed for the MAP-mod concept with 5 viewing angles, where $\Delta \mathrm{DLP} = 0.003$ is assumed. The MAP-mod spectrum for each scenario extends from 385 to 765 nm.

number of viewing angles from 3 to 8, while keeping the spectral range and resolution the same as in section 6.2.1, resulting in 6 instrument setups to evaluate. For each scenario and each setup, $\mathbf{K_{MAP}}$ is computed. $\mathbf{S}_{y,\mathrm{MAP}}$ is fixed, corresponding to

$\Delta I / I$ =3% and $\Delta \mathrm{DLP}$ =0.003. Table 5 lists the viewing angles and the corresponding number of measurements.

The resulting $\langle \Delta \mathrm{XCO}_2 \rangle$ as a function of number of viewing angles is displayed in Figure 4. It shows that $\langle \Delta \mathrm{XCO}_2 \rangle$ drops most sharply from 3 to 4 viewing angles in most cases. Cases 2 and 3 with SZA=60°show also a significant decline in $\langle \Delta \mathrm{XCO}_2 \rangle$ from 4 to 5 viewing angles. Having more viewing angles beyond 5 lowers the aerosol-induced errors only marginally. For instance, having 8 viewing angles does not reduce $\langle \Delta \mathrm{XCO}_2 \rangle$ to 0.6 ppm for the case-1,vegetation,SZA=60°,$\tau$ =0.52 sce-

nario.

**Table 5.** List of the viewing angles in the evaluated MAP-mod setups

| Number of VZAs | VZAs | Total number of measurements |
| --- | --- | --- |
| 3 | 0, ±60 | 288 |
| 4 | ±19, ±57 | 384 |
| 5 | 0, ±40, ±60 | 480 |
| 6 | ±12, ±36, ±60 | 576 |
| 7 | 0, ±20, ±40, ±60 | 672 |
| 8 | ±8, ±25, ±42, ±60 | 768 |

An odd number of viewing angles is preferred over an even number to allow for symmetry and to include a nadir view. Strictly speaking, a minimum of 7 viewing angles are required to have $\langle\Delta XCO_2\rangle$ below 0.6 ppm for all study scenarios. However, 5 viewing angles deliver very similar $\langle\Delta XCO_2\rangle$ and therefore meets our target aerosol-induced error for a vast majority of the study scenarios. Here we adopt 5 viewing angles as the required number of viewing angles.

### 6.2.3 Spectral range

Looking at heritage missions with different spectral coverage, we assess the effect of varying spectral range on the retrieved $XCO_2$. Here we examine four options, i.e. (i) the same spectral range as in sections 6.2.1 and 6.2.2 ('default'), (ii) the default range including UV wavelengths down to 350nm ('with UV'), (iii) the default range without wavelengths shorter than 490 nm ('no UV'), (iv) the default range with 2 additional SWIR wavelengths 1640 and 2250 nm, at which both radiance and DLP measurements are taken ('with SWIR') . For each option, we assume 5 viewing angles at 0,±40,±60 degrees. For setups (i), (ii), and (iii), the spectral resolutions for radiance and DLP are the same as those adopted in section 6.2.1, and the uncertainties are fixed to $\Delta I/I$ =3% and $\Delta DLP$ =0.003. Table 6 summarizes the four setups.

**Table 6.** List of the 4 options for the MAP-mod spectral range

| Setup | Spectral range | Number of radiance measurements | Number of DLP measurements | Total number of measurements |
| --- | --- | --- | --- | --- |
| default | 385-765 | 77 | 19 | 480 |
| with UV | 350-765 | 84 | 22 | 530 |
| no UV | 490-765 | 56 | 12 | 340 |
| with SWIR | 385-2250 | 79 | 21 | 500 |

Figure 5 compares $\langle\Delta XCO_2\rangle$ for the four spectral range options as a function of total aerosol optical depth. Compared to the default setup, it is apparent that adding more measurements in the UV leads to little gain in performance and that removing UV measurements altogether results in a considerably higher $\langle\Delta XCO_2\rangle$ . Without UV measurements, half of the scenarios fail to meet the target 0.6 ppm; in some case-3,vegetation scenarios, $\langle\Delta XCO_2\rangle$ even exceeds 1 ppm. The contribution of the

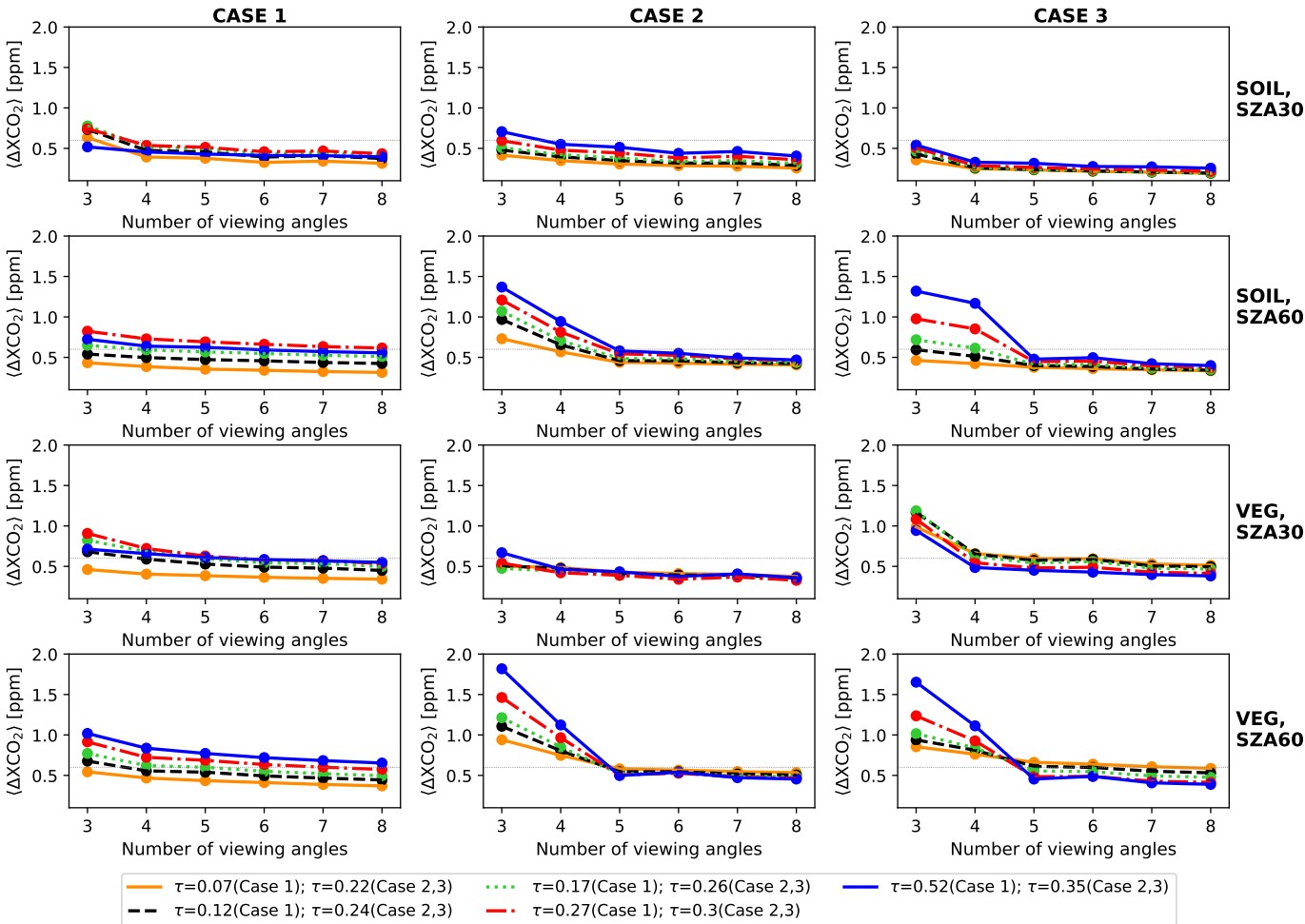

**Figure 4.** $\langle \Delta \mathrm{XCO}_2 \rangle$ as a function of number of viewing angles for all 60 study scenarios. The 5 lines in each panel represent different values of total aerosol optical depth at 765 nm. $\langle \Delta \mathrm{XCO}_2 \rangle$ is computed for the MAP-mod concept with $\Delta I / I$ =3% and $\Delta$DLP =0.003. The MAP-mod spectrum for each scenario extends from 385 to 765 nm.

two SWIR channels, as compared to the default spectral range, is most significant for case 1 scenarios in which a drop of $\langle \Delta \mathrm{XCO}_2 \rangle$ up to $\sim 0.3$ ppm can be seen. For the other scenarios, the SWIR channels have only marginal effects. Given that for case 1 the default spectral range is already very close to the requirement, we conclude that the 'default' range (385-765 nm) is
sufficient.

Following the assessment above, we adopt the default setup as the MAP-mod baseline setup. For clarity, we summarize this setup in Table 7.

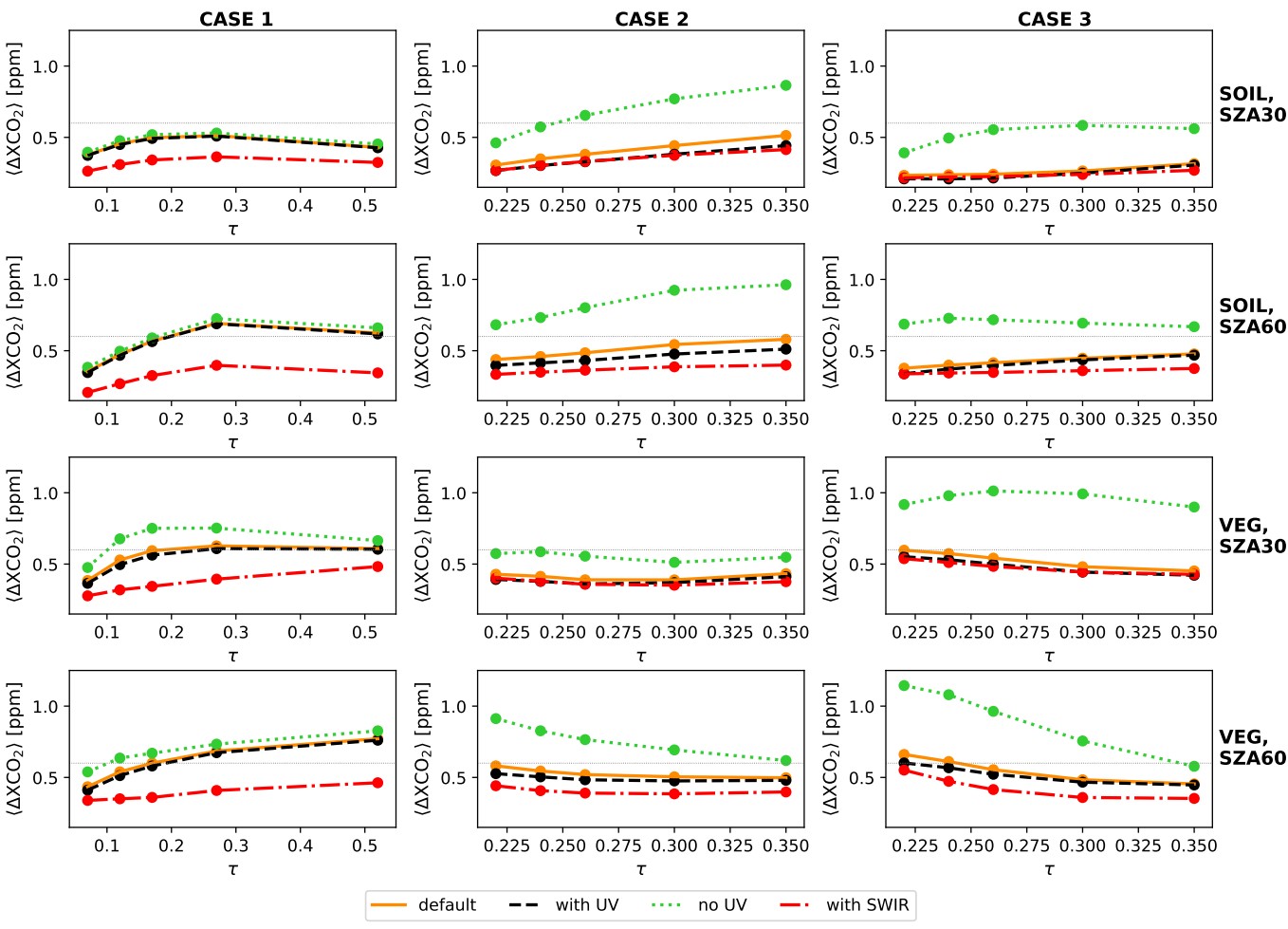

**Figure 5.** $\langle \Delta XCO_2 \rangle$ as a function of total aerosol optical depth at 765nm. The 4 lines in each panel represent 4 possible setups with different spectral range. $\langle \Delta XCO_2 \rangle$ is computed for the MAP-mod concept with 5 viewing angles, $\Delta I / I$ =3% and $\Delta DLP$ =0.003.

### 6.3 MAP-band instrument

For the MAP-band instrument, we consider 6 spectral bands from 410 to 865 nm, close to the 3MI VNIR polarized bands
(Fougnie et al., 2018), and we perform requirement analyses similar to those for MAP-mod. The study of Hasekamp and Landgraf (2007) suggests that there is a strong overlap in angular and spectral information for aerosol retrieval, i.e. as long as the total number of measurements is the same and there are at least 5 viewing angles, instruments with different number of angles and wavelengths yield similar retrieval capability. This implies that the MAP band instrument with 40 viewing angles can deliver a similar performance to MAP-mod, because the total number of measurements of the two concepts would then be
the same. *The details of our proposed baseline setup for the MAP-band concept is provided in Table 8.*

**Table 7.** MAP-mod baseline setup

| Features | Baseline setup |
|---|---|
| Number of VZAs | 5 |
| Viewing angles [degrees] | $0, \pm 40, \pm 60$ |
| Spectral range | 385-765 nm |
| Radiance spectral resolution | 5 nm |
| DLP spectral resolution | 15 at 395nm, 30 at 765nm |
| Number of radiance measurements | 77 |
| Number of DLP measurements | 19 |
| Total number of measurements | 480 |
| Radiance uncertainty | 3% |
| DLP uncertainty | 0.003 |

Figure 6 displays the performance of a MAP-band setup (40 viewing angles, 6 spectral bands, $\Delta I/I$ =3%, $\Delta$DLP =0.003) and the MAP-mod baseline setup. For most scenarios, $\langle \Delta \mathrm{XCO_2} \rangle$ delivered by both concepts are comparable, confirming what is suggested in Hasekamp and Landgraf (2007). MAP-band fares somewhat poorly for scenarios with vegetation surface and SZA=60°in cases 2 and 3. To maintain aerosol-induced errors at 0.6 ppm or lower for these scenarios, we find that the MAP-band measurement uncertainties have to be unfeasibly low, i.e. $\Delta I/I$ must be 1% or less for a $\Delta$DLP of 0.003, or alternatively, $\Delta$DLP must be less than 0.002 for a $\Delta I/I$ of 3%. ~~The details of our proposed baseline setup for the MAP-band concept is provided in Table 8.~~

## 7  XCO₂ retrieval using MAP and CO2M spectrometer measurements

The linear error analysis provides reliable XCO$_2$ error estimates assuming that the inversion problem has been succesfully solved and the global minimum has been found. However, actual retrievals may have difficulties in achieving that and the minimization procedure may get trapped in a local minimum. In this case, the real performance of the iterative retrievals would be worse than that expected from the linear error analysis. For this reason, here we evaluate the retrieval capability of the combined spectrometer and MAP measurements using a full iterative approach (described in section 3.1) on a more diverse set of synthetic scenes. We adopt the baseline ~~MAP-mod setup to simulate MAP observations~~ *setup of the MAP-mod and MAP-band concepts* and we consider the ensemble of 500 simulated scenes (as outlined in section 4) for this joint retrieval exercise.

We apply the same $\chi^2$ filtering as in section 5 to the joint retrieval results. ~~Out of the 500 scenes, 349 (70%) with $\chi^2 \leq 1.5$ are left and their $\Delta$XCO$_2$ are evaluated.~~ $\Delta$XCO$_2$ *is evaluated only in the case of convergence, i.e.* $\chi^2 \leq 1.5$. *For the spectrometer and MAP-mod joint (joint-mod) retrievals, 349 out of the 500 scenes (70%) fulfill this criterion, while for spectrometer and MAP-band combination (joint-band) the number stands at 390 (78%).* Higher convergence rates can potentially be achieved

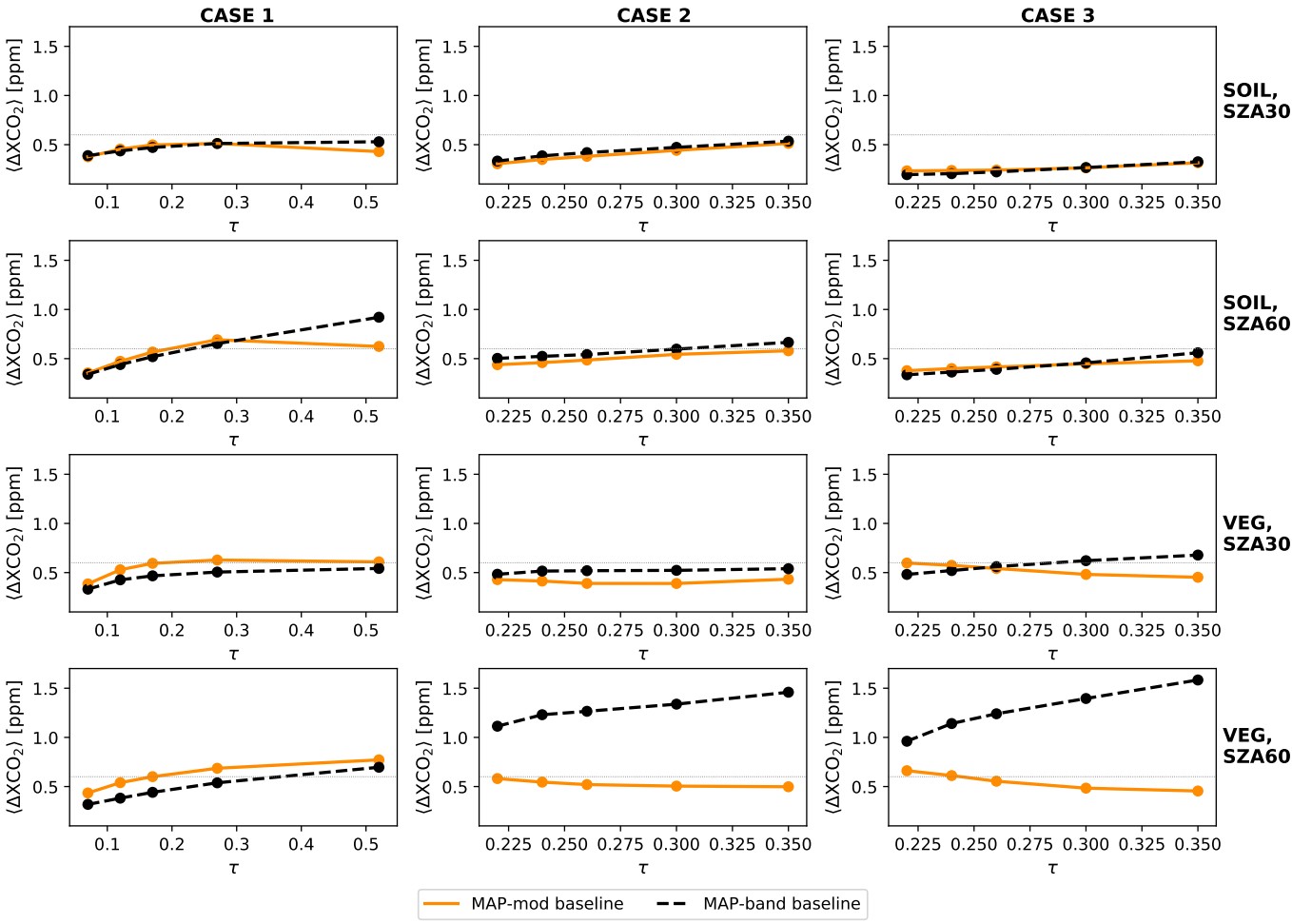

**Figure 6.** $\langle\Delta\mathrm{XCO_2}\rangle$ as a function of total aerosol optical depth at 765nm. The two lines in each panel represent the baseline setups of MAP-mod (orange, solid) and MAP-band (black, dashed).

with further refinements of the iterative scheme and a better approach to select the first guess state vector. As in section 5, $\Delta\mathrm{XCO_2}$ is the difference between the retrieved and the true $\mathrm{XCO_2}$ . Since random instrument errors are added to the simulated spectra, $\Delta\mathrm{XCO_2}$ here is a combination of aerosol-induced errors (which include MAP instrument noise) and spectrometer-noise-induced errors.

Figure 7 plots $\Delta\mathrm{XCO_2}$ for the 349 converged *joint-mod* retrievals as a function of the true values of aerosol optical depth and height, SZA, and albedo. *The same plots for the joint-band retrievals are displayed in Figure 8. Between the two figures, no significant difference is seen with respect to the statistical distribution of* $\Delta\mathrm{XCO_2}$ . ~~The range of $\Delta\mathrm{XCO_2}$ values along the y-axis is kept the same as in Figure 1 for easy comparison. A few things that stand out from comparing the two figures are the substantial reduction in $\Delta\mathrm{XCO_2}$ scatter and the absence of strong $\mathrm{XCO_2}$ bias in the joint retrieval results.~~ Excluding a few

**Table 8.** MAP-band baseline setup

| Features | Baseline setup |
|---|---|
| Number of VZAs | 40 |
| Viewing angles [degrees] | $\pm2, \pm5, \pm8, \pm11, ..., \pm54, \pm57, \pm60$ |
| Wavelengths [nm]$^\dagger$ | 410, 440, 490, 550, 669.9, 863.4 |
| Number of radiance measurements | 240 |
| Number of DLP measurements | 240 |
| Total number of measurements | 480 |
| Radiance uncertainty | 3% |
| DLP uncertainty | 0.003 |

$^\dagger$ In the official baseline setup for the MAP-band instrument, the list of spectral bands includes 753 nm. The channel is added for calibration purposes, owing to its overlap with the NIR band of the CO2M spectrometer. Only radiance measurements are taken in this channel, therefore we do not consider it in our analysis.

obvious outliers, the general trends are as follows. As the aerosol optical depth gets higher, the scatter of $\Delta XCO_2$ increases only mildly with no signs of significant overestimation or underestimation of the retrieved $XCO_2$ . With respect to the aerosol altitude, $\Delta XCO_2$ scatter is practically unchanged with almost no bias visible at any height. As for the SZA, a slightly larger scatter of $\Delta XCO_2$ is observed starting SZA$\sim 60°$with no particular collective offset across the whole range of SZA. For the albedo, the scatter of $\Delta XCO_2$ is maintained over the range considered here.

*The key outcome of this exercise is the stark contrast in the* $\Delta XCO_2$ *distribution between the joint, regardless of the MAP-concept, and the spectrometer-only retrievals in Figure 1. There is a substantial reduction in* $\Delta XCO_2$ *scatter and an absence of strong* $XCO_2$ *bias when MAP measurements are included in the retrievals.* As done for the spectrometer-only retrieval results, we again choose to adopt the median and $P_{SD}$ as a measure of the retrieval bias and the spread of $\Delta XCO_2$ distribution. From the 349 converged *joint-mod* retrievals, the bias is -0.004 ppm and the spread is 0.54 ppm. ~~The latter is~~ *From the 390 converged*

*joint-band retrievals, the bias and* $P_{SD}$ *are 0.02 and 0.52 ppm, respectively. Both* $P_{SD}$ *values from joint-mod and joint-band are* consistent with what we expect from the linear error analysis for the MAP~~mod~~ baseline setup~~s~~ (see ~~e.g.~~ Figure 6). More importantly, ~~0.54 ppm~~ *they* lie~~s~~ well within the CO2M total error budget (0.86 ppm) and therefore compliant with the mission requirements. *Given the similar performance, either one of the joint-mod or joint-band setup is suitable for the CO2M mission.*

Compared to the statistics of the spectrometer-only retrievals, the joint retrieval results obviously represent a major improve-

ment in the accuracy and precision of the retrieved $XCO_2$ . The retrieval bias is reduced by ~~more than two orders of magnitude~~ *at least a factor of 50* and the scatter is lowered by almost a factor of 4. The smaller bias and scatter imply that a higher number of observed scenes can be processed into reliable estimates of $XCO_2$ . Moreover, the absence of $\Delta XCO_2$ correlations with aerosol optical depth, aerosol height, and SZA means there is a minimal risk of regional biases in the L2-products, driven by

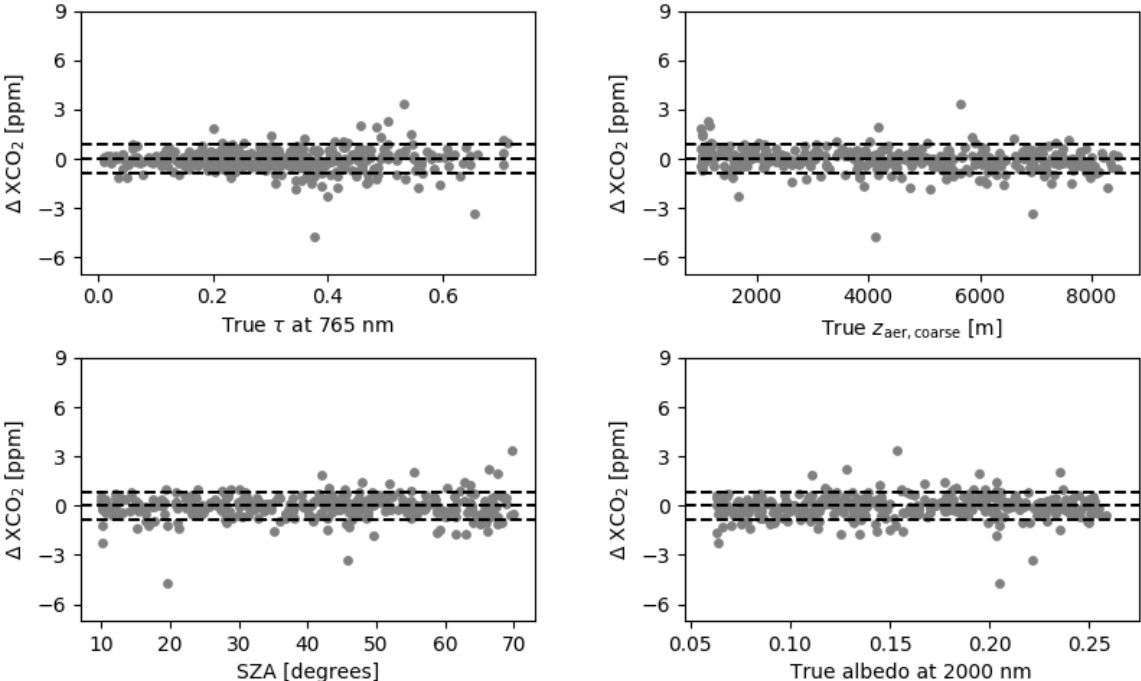

**Figure 7.** Residual $XCO_2$ from the converged joint retrievals using spectrometer and MAP-mod measurements, shown as a function of total aerosol optical depth $\tau$, coarse-mode aerosol height, SZA, and albedo. The input spectra are generated according to the ensemble of synthetic scenes (Section 4). The three dashed horizontal lines indicate $\Delta XCO_2$ at -0.86, 0, and 0.86 ppm.

variations in aerosol properties and SZA. Altogether, such improvements will lead to a higher data yield, better global coverage and a more comprehensive determination of $CO_2$ sinks and sources.

Deployment of a MAP instrument would additionally offer a better insight into the surface reflection properties, which are important factors in simulating the radiation at the top of atmosphere, especially for retrievals over land. In section 5, the simulated spectrometer spectra and the eventual retrieval are based on a Lambertian description of the surface. In reality, the Lambertian assumption does not hold and this would likely lower the $XCO_2$ accuracy of the spectrometer-only retrievals even further, particularly for cases with larger aerosol optical depth because of multiple light scattering between the surface and aerosol particles. *In line with the CO2M mission priority, this work focuses on land surfaces and does not address water bodies or glint measurements. For the glint geometry, direct light dominates the light path distribution so we expect less atmospheric scattering and less aerosol interference. Glint-mode performance with the MAP instrument on board CO2M will be the topic of future research.*

*It should be noted that besides the aerosol-induced errors studied here, there are also other error sources that affect the final performance, most notably due to imperfect spectroscopy (Miller et al., 2005; Hobbs et al., 2020). Such errors can be reduced by improved laboratory measurements of gas spectroscopy. From a retrieval experiment point of view, past studies have shown*

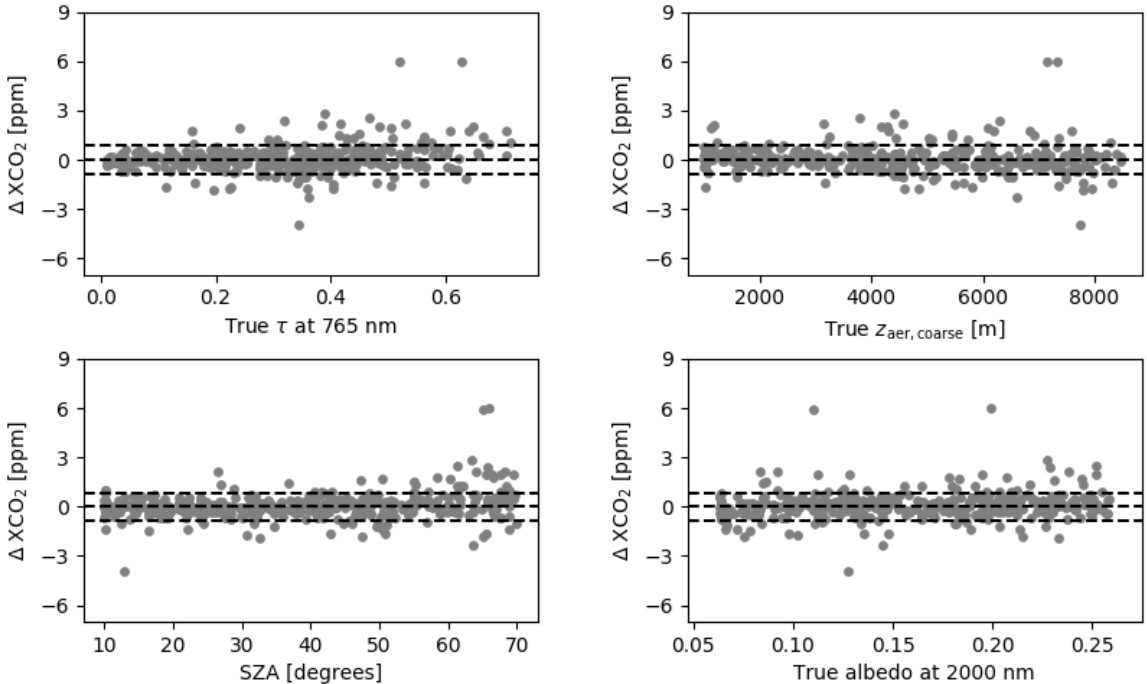

**Figure 8.** Same as Figure 7, but for the converged joint retrievals using spectrometer and MAP-band measurements.

*that retrieval performance on synthetic data, with focus on aerosol-induced errors (Butz et al., 2009) reflect quite well the actual performance on real data from GOSAT and OCO-2 (Guerlet et al., 2013; Wu et al., 2018, 2020).*

## 8 Summary

In the context of ESA's CO2M mission, we investigated the need for an aerosol-dedicated instrument (multi-angle polarimeter or MAP) in support of the CO2M spectrometer to achieve the required $XCO_2$ accuracy and precision. We estimated aerosol-induced $XCO_2$ errors from two $XCO_2$ retrieval approaches on an ensemble of 500 synthetic scenes over land. The first approach represents a ~~customary~~ *custom* way to account for aerosol effects on the retrieved $XCO_2$ using only measurements of a 3-band spectrometer. The second strategy incorporates MAP and spectrometer measurements in a synergistic way to retrieve $XCO_2$ (joint retrieval).

In the ensemble of synthetic scenes, aerosol size distribution is described by a bimodal lognormal function, where each mode follows a Gaussian height distribution. The trace gas total column, aerosol and surface properties, and the solar zenith angle are randomly varied within certain limits to generate 500 atmospheric and geophysical scenes.

For the standard retrieval exercise using only spectrometer data, we employed the RemoTeC algorithm that has been widely used for greenhouse gas retrievals from space. In RemoTeC, a simple aerosol model is used, i.e. aerosol size distribution is retrieved following a monomodal power-law parametrization. Out of 500 retrievals, 69% meet our $\chi^2$ convergence criterion.

The median value of the residual $XCO_2$ ($\Delta XCO_2$) from the converged retrievals is 1.12 ppm and the spread is 2.07 ppm. Given that the total $XCO_2$ error budget of the CO2M mission is 0.86 ppm (the quadratic sum of the required $XCO_2$ accuracy of 0.5 ppm and the required $XCO_2$ precision of 0.7 ppm), the results show that the standard retrieval approach is greatly inadequate and does not comply with the mission requirements. Furthermore, the retrieval performance is markedly degraded at high aerosol optical depth, high aerosol altitude and low SZA. This may lead to biases in determining $CO_2$ emissions from polluted areas where $CO_2$ and aerosols are co-emitted.

Prior to performing the joint retrieval, we conducted a requirement analysis to construct a baseline setup for each of the two alternative MAP concepts being considered for the CO2M mission, i.e. MAP-mod and MAP-band. The MAP-mod concept is based on a spectral modulation technique where polarization information is encoded in the modulation pattern of the radiance spectrum, while the MAP-band instrument acquires radiance and polarization measurements at specific discrete spectral bands. The MAP-mod instrument inherits from SPEXone and the MAP-band wavelength channels inherit from 3MI polarized VNIR bands. The optimal baseline setups for the MAP-mod and for the MAP-band instrument designs are found through a linear error analysis that is formulated to mimic a joint retrieval. In particular, we investigated three aspects of a MAP instruments, i.e. the measurement uncertainties, number of viewing angles, and wavelength range. For the MAP-mod concept, the baseline setup includes 5 viewing angles ($\pm60°,\pm40°,0°$), 77 radiance measurements (with $\Delta I/I$ =3%) and 19 DLP measurements (with $\Delta DLP$ =0.003) ranging from 385 nm to 765 nm. The baseline setup for the MAP-band concept requires 40 viewing angles (from -60°to 60°), 6 spectral bands between 410 and 865 nm at which both radiances and DLP are measured, with the same radiometric and polarimetric uncertainty requirements as for the MAP-mod baseline. The baseline setups of MAP-mod and MAP-band have generally similar performance.

To implement the joint retrieval, we further developed an existing aerosol retrieval algorithm to include features related to the spectrometer measurements and to the derivation of trace gas total columns. ~~Using this retrieval tool~~*With this tool and using the combined spectrometer and MAP-mod (MAP-band) measurements,* 70% *(78%)* of the 500 retrievals reach convergence according to our $\chi^2$ criterion. Of the converged ones, the median $\Delta XCO_2$ is found at -0.004 *(0.02)* ppm, and the spread stands at 0.54 *(0.52)* ppm, ~~which is~~ consistent with what is expected from the linear error analysis for the MAP-mod *(MAP-band)* baseline setup. More importantly, 0.54 *(0.52)* ppm fits well within the CO2M $XCO_2$ error budget of 0.86 ppm and therefore is compliant with the CO2M requirements. There is not any appreciable correlation in our test ensemble between $\Delta XCO_2$ with the aerosol optical depth, aerosol height, solar zenith angle, or albedo.

The results of the joint retrieval *(for either of the two MAP concepts)* represent a significant improvement in the retrieved $XCO_2$ accuracy with respect to the standard retrieval approach using a 3-band spectrometer only. The bias and the scatter of $\Delta XCO_2$ are much smaller in the joint retrieval, which would ultimately translate to better estimates of $CO_2$ sinks and sources. Figure 9 concisely sums up the main results of this study. It shows ~~to a large extent~~ the benefit of having MAP observations to support $XCO_2$ retrieval. It shows that MAP observations are indispensable in minimizing aerosol-induced $XCO_2$ errors and in achieving the $XCO_2$ precision and accuracy required by the CO2M mission.

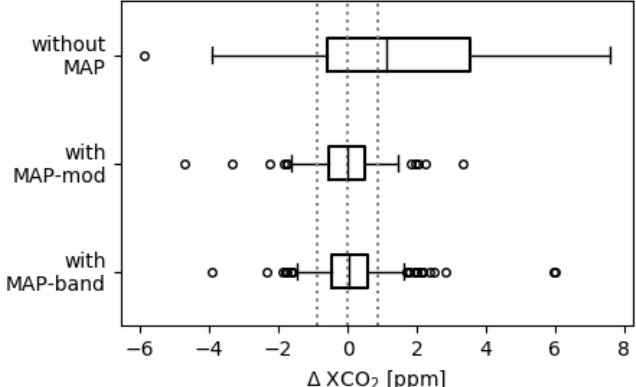

**Figure 9.** Boxplot of $\Delta XCO_2$ from retrievals on the synthetic scenes using 3-band spectrometer measurements only ('without MAP') versus those from retrievals using the combined MAP-mod and spectrometer data ('with MAP-mod'), *and those from the combined MAP-band and spectrometer data ('with MAP-band')*. The rectangles are outlined by the 15.9th and the 84.1th percentiles (P(15.9) and P(84.1)) with the 50th percentile (median) indicated by a vertical line inside the rectangles. The length of the whiskers are set to the difference between P(84.1) and P(15.9). The circles indicate retrievals with $\Delta XCO_2$ beyond the extent of the whiskers. The three dotted vertical lines mark -0.86, 0, and 0.86 ppm.

## Appendix A: Lognormal distribution

The lognormal distribution is used in this paper to describe the size distribution $n(r)$ of each aerosol mode. It reads as follows:

$$n(r) = \frac{1}{r \ln s_g \sqrt{2\pi}} \exp\left[-\frac{(\ln r - \ln r_m)^2}{2(\ln s_g)^2}\right], \tag{A1}$$

with $r$, $r_m$ and $s_g$ being radius, median radius and the geometric standard deviation. Here, we use effective radius $r_{\text{eff}}$ and effective variance $v_{\text{eff}}$ in place of $r_m$ and $s_g$, where

$$v_{\text{eff}} = \exp((\ln s_g)^2) - 1, \tag{A2}$$
$$r_{\text{eff}} = r_m (1 + v_{\text{eff}})^{2.5}. \tag{A3}$$

**Appendix B: Retrieved aerosol properties**

In all of our retrieval exercises, aerosol properties are fitted alongside $XCO_2$ and some surface parameters. Here we discuss the retrieved aerosol properties in our experiments. In the main body of the paper, we compare the $XCO_2$ retrieval performance of the joint MAP-spectrometer and the spectrometer-only setup. However, it is interesting to also look into the retrieval performance with respect to the inferred aerosol properties. For this reason, we carry out the classical MAP-only retrieval using the

aerosol retrieval algorithm (Hasekamp et al., 2011; Fu and Hasekamp, 2018; Fu et al., 2020), from which the joint retrieval tool originates. This means the forward model, state variables, and the inversion procedure described in section 3.1 also apply here, only without trace gases in the state vector and without spectrometer-related aspects. As input, the MAP-only retrieval uses the same MAP synthetic measurement as in the joint retrieval (section 4; Table 3). Given the performance similarity between MAP-mod and MAP-band (Figures 6,7,8), we arbitrarily choose to adopt the MAP-mod concept (Table 7) for this exercise.

We begin by comparing aerosol optical depth estimates from the three retrieval approaches, i.e spectrometer-only, MAP-only, and the joint MAP(mod)-spectrometer setups. Because spectrometer-only retrieval relies on a simpler aerosol parameterization, the total $\tau$ is the only aerosol property from the 3 retrieval types that we can directly compare. Figure B1 shows scattter plots of $\tau$ at 765 nm for the spectrometer-only, MAP-only, and joint retrievals with the corresponding RMSE provided. For the MAP-only and joint retrievals, the $\tau$ constitutes the sum of the fine- and the coarse-mode optical depths. Only the converged retrievals
are included in the figure, i.e. 343, 349, and 495 data points from the spectrometer-only, joint, and MAP-only retrievals, respectively. It is plain to see that $\tau$ retrieval is severely compromised in the absence of MAP data. Compared to MAP-only retrieval, having spectrometer measurements on top of MAP data decreases the RMSE.

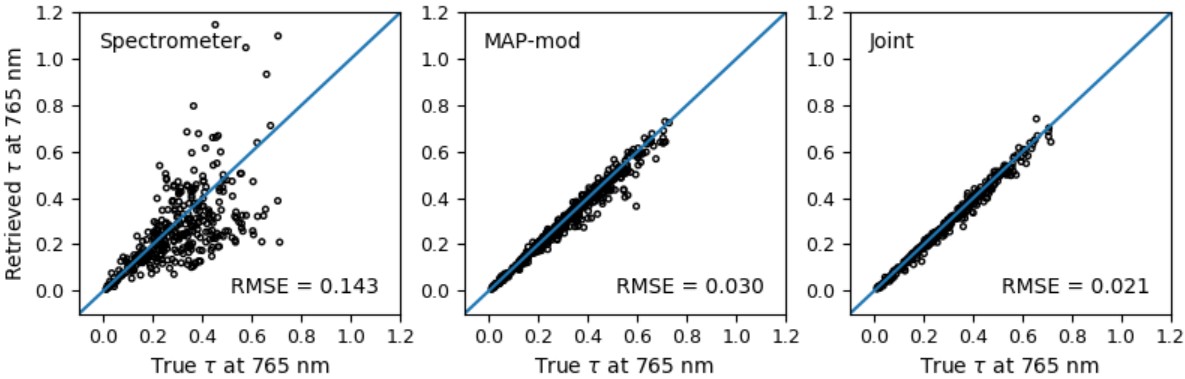

**Figure B1.** Aerosol optical thickness from spectrometer only (left), MAP-only (middle), and joint (right) retrievals compared to the true values. The diagonal solid lines outline the one-to-one correspondence.

To help us gain a better insight into the overall retrieval system, Figures B2 and B3 display the MAP-only and the joint
retrieval performance for each aerosol property in the state vector (Table 2). As in the previous figure, The RMSE is given in every panel in the right bottom corner. Both retrieval appraoches exhibit similar performance, where aerosol properties are generally well retrieved. On a closer look, the largest performance difference between the two retrieval approaches are found for $r_{\mathrm{eff}}$, $z_{\mathrm{aer}}$, and $\tau$ of the coarse mode. The retrieval of these coarse-mode parameters benefit from the spectrometer measurements because the SWIR spectral range adds sensitivity to coarse-mode aerosols.

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

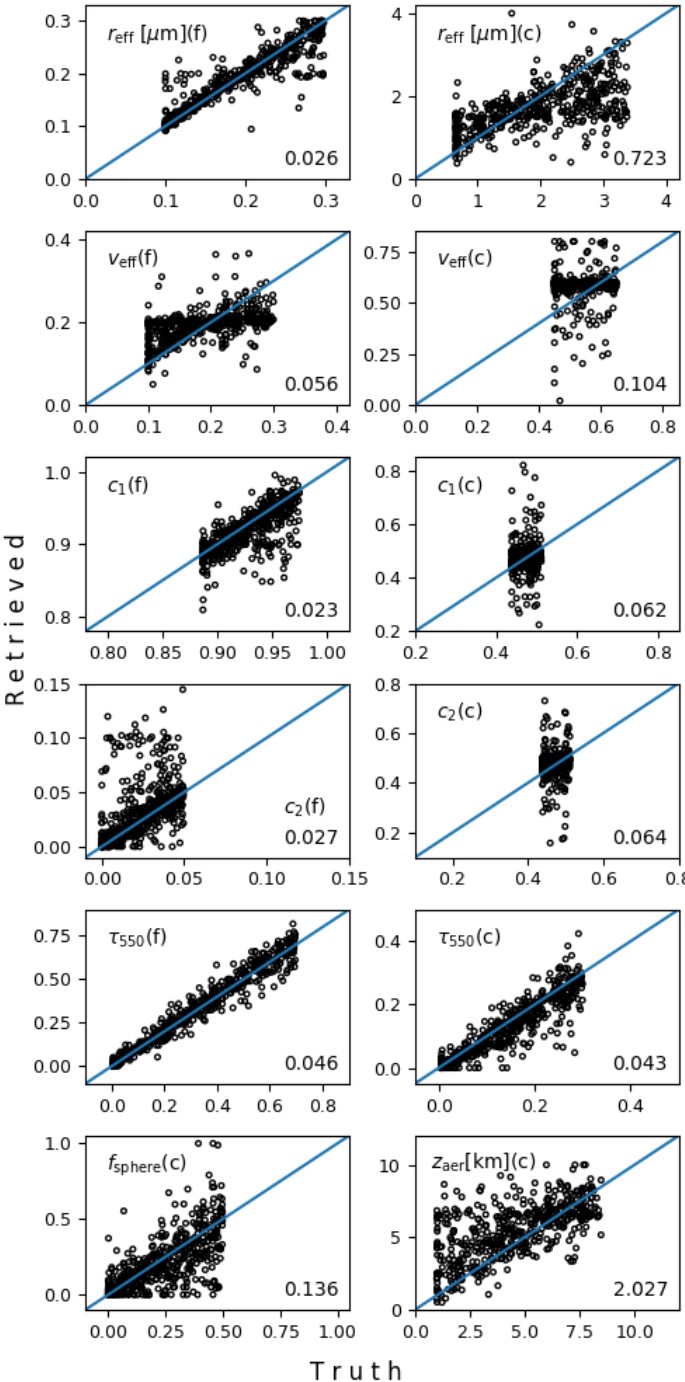

**Figure B2.** Aerosol properties retrieved using MAP-only measurements compared to the truth. In each panel, the letter 'c' or 'f' in parentheses behind the aerosol variable name indicates the aerosol mode, i.e. coarse or fine. The RMSE for each parameter is given in the bottom right corner. $\tau_{550}$ is the aerosol optical depth at 550 nm. The diagonal solid lines outline the one-to-one correspondence.

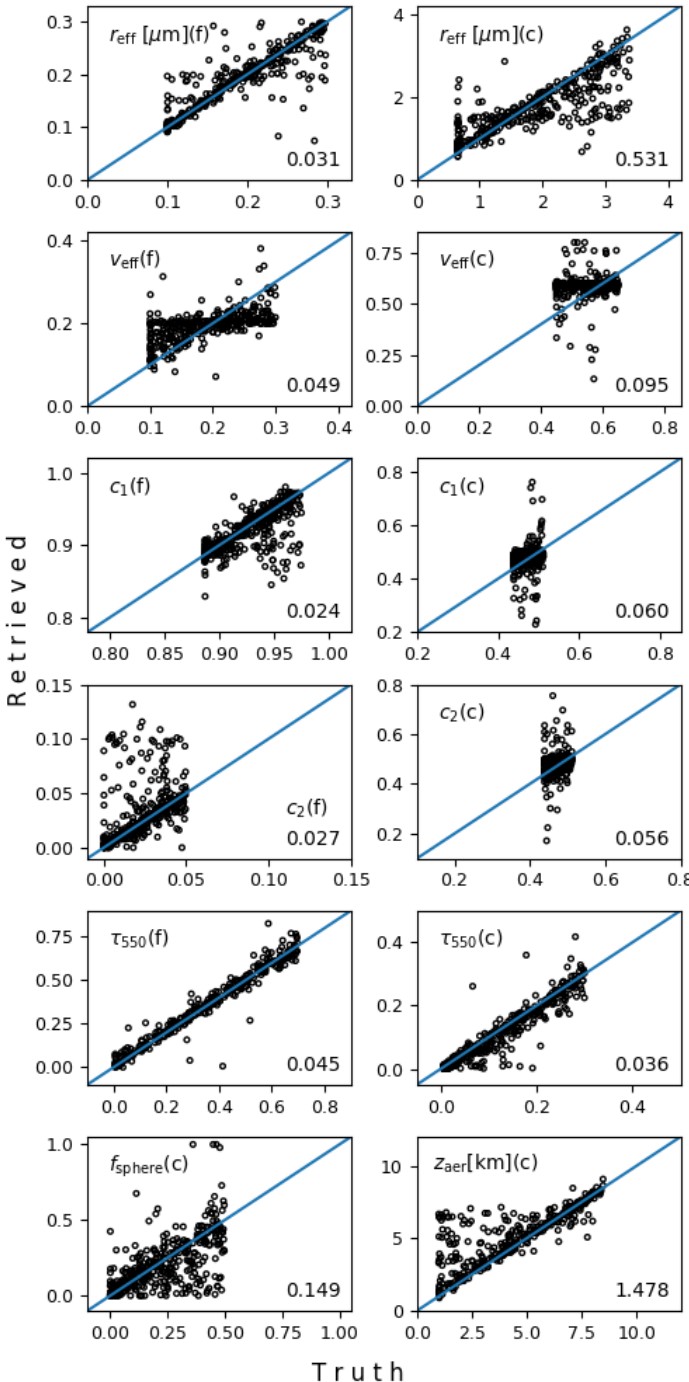

**Figure B3.** Same as Figure B2 but for the joint retrieval.