# Peer review of "Anthropogenic CO2 monitoring satellite mission: the need for multi-angle polarimetric observations"

_Atmospheric Measurement Techniques, 2020_

## Referee Comment (RC1) · Robert Roland Nelson (Referee) · 20 Jul 2020

**General comments:**

The manuscript entitled, "Anthropogenic $CO_2$ monitoring satellite mission: the need for multi-angle polarimetric observations" presents an analysis of the value of adding a multi-angle polarimeter (MAP) to the Copernicus anthropogenic $CO_2$ monitoring (CO2M) mission. Using synthetic observations, they assessed the precision and accuracy of the CO2M XCO2 estimates from a 3-band spectrometer with and without the addition of auxiliary MAP information and found that MAP reduced the errors to below the mission requirements. They also assess the specific technical properties of the desired MAP instrument via linear error analysis. The manuscript is thorough, clear, and very well-written and I recommend publication in AMT after the authors address minor comments below.

**Specific comments:**

- In Section 5, did you compare ΔXCO2 to the retrieved state vector elements (e.g. retrieved AOD)? This can be informative.

- Regarding the errors of the standard retrieval approach, I think it is worth noting that you could certainly reduce those errors to within the mission requirements, but only by heavily post-filtering (and maybe bias correcting) the data. This new joint retrieval means you can keep significantly more data!

- P2 L36: So the revisit time is daily for 40S to 40N?

- P4 L111 "*in this work we are interested primarily in XCO2 and do not discuss the retrieved aerosol properties.*"

It would be interesting to at least look at how well MAP+CO2M does at retrieving the aerosol properties, compared to either MAP or CO2M alone. Does the aerosol information in the O2 and SCO2 bands add anything useful to the MAP aerosol results?

- P6 L168 "*We take the input vertical profiles of the trace gases as a given and retrieve the total columns via scaling factors.*"

Where do you get the prior profiles from?

- P6 L170 "*there are only 4 aerosol parameters that are not retrieved, i.e. $f_{sphere}$, $z_{aer}$ of the fine-mode aerosol, and $w_{aer}$ for both modes.*"

I understand that the radiances aren't very sensitive to $w_{aer}$, but why aren't $f_{sphere}$ and $z_{aer}$ of the fine-mode aerosol solved for? Fixing these parameters to the truth obviously becomes problematic if real measurements end up being sensitive to them.

- P8 L212 *"the error analysis follows a two-step approach"*

Could you briefly explain the reasoning for splitting it into two steps?

- Figure 1 and corresponding text: make it clear when you are talking about the true scene properties vs. the retrieved properties.

- P14 L357 *"With a $P_{SD}$ above 2 ppm, $XCO_2$ retrievals based on only spectrometer measurements do not meet the mission requirements by a very wide margin (note that we do not apply postretrieval filtering here)."*

Can you comment on how much you might expect $P_{SD}$ to be reduced with post-filtering in these simulations?

- P17 L415 *"Given that the improvement in $\Delta I=I$ from 3% to 2% is a major technical challenge…"*

Would reducing $\Delta DLP$ to 0.002 be easier?

- Figure 5: why does case 1 benefit so much from the SWIR bands?

- P21 L465 *"The details of our proposed baseline setup for the MAP-band concept is provided in Table 8."*

Maybe put this line earlier in Section 6.3.

- P25 L537 *"The baseline setups of MAP-mod and MAP-band have generally similar performance."*

But do the few scenarios where the MAP-band instrument does really poorly mean that MAP-mod is the better choice? Briefly discuss if you recommend one MAP instrument over the other, or if either would be acceptable.

**Technical comments:**

P2 L50: change to "shorten or lengthen"

P3 L68: change to "It clearly shows the benefit…"

P5 L135 change to "spectrometer"

P16 L406 looks like the highest value in Fig. 2 is actually just above 2.5, not ~2.4.

P24 L511 change to "custom"

P26 L549 remove "to a large extent"

---

## Referee Comment (RC2) · Anonymous Referee #3 · 22 Jul 2020

**General Comments:**

The manuscript by Rusli et al. with the title *"Anthropogenic CO2 monitoring satellite mission: the need for multi-angle polarimetric observations"* investigates the added value of multi-angle polarimeter (MAP) measurements in the context of the Copernicus candidate mission for anthropogenic CO2 monitoring (CO2M).

Scattering by aerosols and cirrus are the main sources of uncertainties in retrieving XCO2 from solar backscattered radiation. Additional MAP observations are expected to provide information about aerosols that are useful for improving XCO2 accuracy. Two different MAP instrument concepts are considered in this analysis: MAP-mod and MAP-band. The authors determine the instrument specifications for both concepts that are required to achieve XCO2 accuracy and precision that align with the requirements of the mission.

Adopting the derived MAP instrument specifications, a retrieval exercise using a spectrometer only and spectrometer + MAP joint retrieval is performed. The study shows that the MAP auxiliary information help to reduce XCO2 errors below mission requirements.

The manuscript is well written and structured in a clear and sensible fashion. It is suitable for publication in AMT after some minor corrections listed below.

**Specific Comments:**

P2 L36: *"The CO2M mission is designed as a constellation of up to 3 satellites with imaging capabilities and a revisit time of 2-3 days for latitudes (poleward of) 40 degrees."*

- What is the revisit time between 40S and 40N?

P2 L39: *"As opposed to currently operational CO2 missions that are designed to observe natural CO2 fluxes, with the exception of OCO-3 (Basilio et al., 2019), the CO2M mission is intended to measure anthropogenic emissions (Pinty et al., 2017)."*

- Earlier in the manuscript you mention that the primary sounders are nadir-looking, very similar to currently operational CO2 missions. What makes the CO2M mission *better* suited to measure anthropogenic emissions compared to other missions?

P5 L142: *"Atmospheric vertical profiles of temperature, H2O, CO2, and CH4 are provided as input."*

- Please specify the source/origin of the vertical profiles.

P6 L168: *"We take the input vertical profiles of the trace gases as a given and retrieve the total columns via scaling factors."*

- See above (please specify the source/origin of the vertical profiles).

P14 L357: *"With a PSD above 2 ppm, XCO2 retrievals based on only spectrometer measurements do not meet the mission requirements by a very wide margin (note that we do not apply post-retrieval filtering here)."*

- How does PSD change after a post-retrieval filtering is applied? Does it help to fulfill mission requirements?

P22 L473: *"We adopt the baseline MAP-mod setup to simulate MAP observations and we consider the ensemble of 500 simulated scenes (as outlined in section 4) for this joint retrieval exercise."*

- Why only for the MAP-mod setup and not also for the MAP-band setup? It would be interesting to see a similar figure to Fig. 7 showing MAP-band results.

**Technical Comments:**

P5 L135: *"spetrometer"* -> spectrometer

P5 L138: *"The measured radiances are simulated by convolving..."*

- The word *"measured"* might cause confusion here.

P12 L312: "simulated measurements"

- It's either simulated or measured, not both.

P16 L406: "〈ΔXCO2〉 *can be as high as ~2.4 ppm for the highest ΔI/I and ΔDLP considered here.*"

- Looks more like ~2.6 ppm to me.

---

## Referee Comment (RC3) · Annmarie Eldering (Referee) · 24 Jul 2020

general comments

This is a very interesting paper that addresses a very important, relevant question. The authors use a simulation system to investigate the benefits of including multiple angles and polarization sensitivity for instrumentation aimed at characterizing XCO2. It is a well structure paper - clearly explains the experiment that was conducted and the outcome. There are a few areas where I'd like to see some additional information, as this work suggests a real transformation in our ability to retrieve XCO2 from space, so any critical assumptions should be articulated.

[Figure]

specific comments 1) The performance shown in Figure 7 is astonishingly good. It would be a significant advancement if we can see this performance with remote sensing data. Therefore, it is critical that there is an understanding of how the simulation relates to actual data. Without experimental data that has the features described, this is hard to do, but I believe that a number of groups have performed simulation and then applied their algorithms to real data - for example RemoteC and ACOS applied to GOSAT and OCO-2? If earlier work informs us how do the simulations compare with the reality we can gain confidence that there are not significant error sources missing from the simulation system. The Wu et al paper on MSR makes a brief comparison between simulations and actual OCO-2 retrievals. I would recommend that the authors review the literature and see if there are more detailed discussions in Butz et al or O'Dell et al. to clarify the expected relationship between simulations of XCO2 retrievals and real life performance.

2) The separation of the spectrometer error out from the other errors is a good strategy. How did you decide on that allocation?

3) You don't have any discussion of the errors that will come from weaknesses in the forward model - the gas spectroscopy has remained a source of error for the OCO-2 mission, and I fully expect this will remain. How can you also consider that error in your analysis or estimate the impact?

4) The Frankenberg et al paper (2012) addresses multiple angle measurements and how they might help both aerosol and xco2 characterization. that paper should be cited in the introduction where a review of literature on the interference of aerosols and the value of multi-angle measurements is presented.

5) Is the the linear error analysis section with all the OE equations really needed? Citation of earlier papers (such as Hasekamp et al or Kulawik et al) should be sufficient. Alternatively, include the central equation in the paper and the rest in the appendix. Don't need to lay out that math in every paper that uses OE and applied linear error

analysis.

6) Is there enough difference in the aerosol variables of the simulation and those in the retrieval?? Some of the terms remain the same, and it isn't clear if that contributes to . I think it would be useful to have one table that has the simulation and retrieval info all in once place. I found myself repeatedly flipping back and forth so I could see how the retrieval set up differed from the simulation set up.

7) I am very interested in the performance of the aerosols themselves. The authors just say this won't be addressed. If this is to be written up in a separate paper, say that. If not, some information about the performance should be included. This could even be an appendix - There is a lot of insight to be gained about the overall retrieval system if we see all of the parameters.

8) What is the variability of water vapor and temperature profile information? Where we they drawn from? Was there any analysis of correlation of errors with these variables?

9) Again, the results presented here are impressive - a significant advance for remote sensing of XCO2. The simulations are all conducted for land surfaces, as the driver for this work is the study of human emission of CO2. But, for the larger carbon cycle science community, such an advance would be important. Can the authors add a few comments about how this work could be extended for glint measurements or if they explored the performance over water bodies (perhaps at a range of distances from the glint spot)? Or perhaps this is planned work for a future manuscript?

technical corrections

1) line 33: spelling of Commission

2) line 73: verb and subject don't match. Also, sentence structure us awkward. Suggest rewording to this "Linear error analysis is part of our study, to derive the optimal instrument specification for each of the two MAP concepts with regard to wavelength range, number of viewing angles and the measurement uncertainties."

[Figure]

3) line 77 and following - I don't think commas are needed. These sentence are correct if written this way: For the retrieval input we generate synthetic measurements that correspond to an ensemble of atmospheric and geophysical scenes over land. The MAP instrument for which the synthetic measurements are generated is tailored to the CO2M mission precision and accuracy requirements.

4) line 382 - refer to Equations A1, 2-4. What is A1? There is no appendix that I am aware of.

---

## Author Comment (AC2) · 30 Oct 2020

**Reply to Anonymous Referee #3**

We thank the referee for their time and effort in critically reading and reviewing our manuscript. Below we reproduce the questions/comments in bold and address them in plain text. Changes in the manuscript are marked in different colors. In response to the overall reviews, we have expanded the manuscript to include
- joint retrieval results in which we adopt the MAP-band concept (section 7), and
- discussion on the retrieved aerosol properties in Appendix B.

**General Comments:**
**The manuscript by Rusli et al. with the title "*Anthropogenic CO2 monitoring satellite mission: the need for multi-angle polarimetric observations*" investigates the added value of multi-angle polarimeter (MAP) measurements in the context of the Copernicus candidate mission for anthropogenic CO2 monitoring (CO2M).**
**Scattering by aerosols and cirrus are the main sources of uncertainties in retrieving XCO2 from solar backscattered radiation. Additional MAP observations are expected to provide information about aerosols that are useful for improving XCO2 accuracy. Two different MAP instrument concepts are considered in this analysis: MAP-mod and MAP- band. The authors determine the instrument specifications for both concepts that are required to achieve XCO2 accuracy and precision that align with the requirements of the mission.**
**Adopting the derived MAP instrument specifications, a retrieval exercise using a spectrometer only and spectrometer + MAP joint retrieval is performed. The study shows that the MAP auxiliary information help to reduce XCO2 errors below mission requirements.**
**The manuscript is well written and structured in a clear and sensible fashion. It is suitable for publication in AMT after some minor corrections listed below.**
Thank you for your positive remarks and careful reading.

**Specific comments:**
**[1] P2 L36: "*The CO2M mission is designed as a constellation of up to 3 satellites with imaging capabilities and a revisit time of 2-3 days for latitudes (poleward of) 40 degrees.*"**
**What is the revisit time between 40S and 40N?**
The global revisit time (including 40S-40N) is 5 days. We adjusted the corresponding sentence in the manuscript to include this information (page 2 lines 38-39).

**[2] P2 L39: "*As opposed to currently operational CO2 missions that are designed to observe natural CO2 fluxes, with the exception of OCO-3***

*(Basilio et al., 2019), the CO2M mission is intended to measure anthropogenic emissions (Pinty et al., 2017)."*
**Earlier in the manuscript you mention that the primary sounders are nadir-looking, very similar to currently operational CO2 missions. What makes the CO2M mission better suited to measure anthropogenic emissions compared to other missions?**
CO2M has an imaging capability, as mentioned in the introduction (line 38), with a large swath (250 km). Combined with a global coverage within a week, this allows us to observe extended emission plumes with a single overpass (e.g. Kuhlmann et al. 2019 - complete reference is available in the manuscript).

[3] **P5 L142: *"Atmospheric vertical profiles of temperature, H2O, CO2, and CH4 are provided as input."***
**Please specify the source/origin of the vertical profiles.**

**P6 L168: *"We take the input vertical profiles of the trace gases as a given and retrieve the total columns via scaling factors."***
**See above (please specify the source/origin of the vertical profiles).**

The atmospheric vertical profiles, as well as the trace gases originate from the AFGL profiles, with CO2 scaled up such that XCO2 = 400 ppm.
We added this information to the manuscript (section 4 page 11 lines 300-302).

[4] **P14 L357: *"With a PSD above 2 ppm, XCO2 retrievals based on only spectrometer measurements do not meet the mission requirements by a very wide margin (note that we do not apply post-retrieval filtering here)."***
**How does PSD change after a post-retrieval filtering is applied? Does it help to fulfill mission requirements?**
When we filter out the converged runs with retrieved aerosol optical depth (at 765 nm) > 0.3, the $P_{SD}$ reduces to 1.95 ppm. Lowering the AOD threshold to 0.2 decreases $P_{SD}$ further to 1.66 ppm. Mission requirements are still not fulfilled after such post-retrieval filtering.
This information has been added to the manuscript (section 5 page 15 lines 376-379).

[5] **P22 L473: *"We adopt the baseline MAP-mod setup to simulate MAP observations and we consider the ensemble of 500 simulated scenes (as outlined in section 4) for this joint retrieval exercise."***
**Why only for the MAP-mod setup and not also for the MAP-band setup? It would be interesting to see a similar figure to Fig. 7 showing MAP-band results.**
We have now performed joint retrievals using the MAP-band baseline setup. The analysis is presented in section 7 which also includes a similar figure to Fig. 7 but for MAP-band results.

**Technical comments:**

**[6] P5 L135: "spetrometer" -> spectrometer**
Corrected (now line 145).

**[7] P5 L138: "The measured radiances are simulated by convolving..."**
**The word "measured" might cause confusion here.**
We removed the word 'measured' (now page 6 line 148).

**[8] P12 L312: "simulated measurements"**
**It's either simulated or measured, not both.**
We replaced "simulated measurements" with "synthetic measurements" (now page 13 line 332).

**[9] P16 L406: "⟨ΔXCO2⟩ can be as high as ~2.4 ppm for the highest ΔI/I and ΔDLP considered here."**
**Looks more like ~2.6 ppm to me.**
It is actually 2.52 ppm. The text now (page 17 line 432) reads:
"⟨ΔXCO2⟩ can be as high as 2.52 ppm for the highest ΔI/I and ΔDLP considered here."

---

## Author Comment (AC1)

**Reply to Robert Roland Nelson**

We thank the referee for his time and effort in critically reading and reviewing our manuscript. Below we reproduce the questions/comments in bold and address them in plain text. Changes in the manuscript are marked in different colors. In response to the overall reviews, we have expanded the manuscript to include
- joint retrieval results in which we adopt the MAP-band concept (section 7), and
- discussion on the retrieved aerosol properties in Appendix B.

**General comments:**
**The manuscript entitled, "Anthropogenic $CO_2$ monitoring satellite mission: the need for multi-angle polarimetric observations" presents an analysis of the value of adding a multi-angle polarimeter (MAP) to the Copernicus anthropogenic $CO_2$ monitoring (CO2M) mission. Using synthetic observations, they assessed the precision and accuracy of the CO2M XCO2 estimates from a 3-band spectrometer with and without the addition of auxiliary MAP information and found that MAP reduced the errors to below the mission requirements. They also assess the specific technical properties of the desired MAP instrument via linear error analysis. The manuscript is thorough, clear, and very well-written and I recommend publication in AMT after the authors address minor comments below.**

Thank you for your positive remarks and careful reading.

**Specific comments:**

**[1] In Section 5, did you compare ΔXCO2 to the retrieved state vector elements (e.g. retrieved AOD)? This can be informative.**

In Figure 1 in section 5, we show ΔXCO2 as a function of true AOD, aerosol height, albedo, and SZA. Below we show the same kind of plots, but as a function of the retrieved AOT, aerosol height, and albedo for your comparison. Please note that spectrometer-only retrieval considers only one mode of aerosols so a fair comparison between the retrieved $z_{aer}$ and the true $z_{aer,coarse}$ is difficult. Retrieved AOD tends to cluster at smaller values than the truth (as also seen in Figure B1 in appendix B). Other than that, we believe that the retrieved values present no additional information. The scatter and the overall trend are similar to the true AOD - ΔXCO2 plot in Fig. 1, which is also the case for the albedo plot.

A statement about this has been added to the manuscript (section 5, page 13 lines 346-347).

[Figure]

**[2]** **Regarding the errors of the standard retrieval approach, I think it is worth noting that you could certainly reduce those errors to within the mission requirements, but only by heavily post-filtering (and maybe bias correcting) the data. This new joint retrieval means you can keep significantly more data!**
Indeed, this is now clarified in section 5 (page 15 lines 383-385).

**[3]** **P2 L36: So the revisit time is daily for 40S to 40N?**
The global revisit time (including 40S-40N) is 5 days. We adjusted the corresponding sentence in the manuscript to include this information (page 2 lines 38-39).

**[4]** **P4 L111 *"in this work we are interested primarily in XCO2 and do not discuss the retrieved aerosol properties."***
**It would be interesting to at least look at how well MAP+CO2M does at retrieving the aerosol properties, compared to either MAP or CO2M alone. Does the aerosol information in the O2 and SCO2 bands add anything useful to the MAP aerosol results?**
To answer this question we performed MAP-only retrievals and made comparisons of the retrieved aerosol properties in Appendix B. Please consult that section for details. In general, the results show significant improvement in the retrieved aerosol properties when MAP measurements are available. The comparison between MAP-only and joint retrieval results indicates smaller retrieval errors for the effective radius, the height and the optical depth of the coarse-mode aerosols when spectrometer measurements are included.

**[5] P6 L168 "*We take the input vertical profiles of the trace gases as a given and retrieve the total columns via scaling factors.*"**
**Where do you get the prior profiles from?**
The trace gas vertical profiles in the simulation are taken from the AFGL atmospheric profiles, with CO2 profile is scaled such that XCO2 = 400 ppm. This information is now included in the manuscript (section 4 page 11 lines 300-302).

**[6] P6 L170 "*there are only 4 aerosol parameters that are not retrieved, i.e. $f_{sphere}$, $z_{aer}$ of the fine-mode aerosol, and $w_{aer}$ for both modes.*"**
**I understand that the radiances aren't very sensitive to $w_{aer}$, but why aren't $f_{sphere}$ and $z_{aer}$ of the fine-mode aerosol solved for? Fixing these parameters to the truth obviously becomes problematic if real measurements end up being sensitive to them.**
The choice to fit $f_{sphere}$ only for the coarse mode has been made because non-spherical particles mostly relate to mineral dust which is predominantly in the coarse mode. For $z_{aer}$ our choice would be logical for a situation with industrial aerosol (fine mode) in the boundary layer and an elevated dust layer (coarse mode). The reviewer is correct that this choice may not be optimal. In particular, elevated layers of fine mode aerosols will be present in biomass burning plumes. It may be a better choice to fit one value for $z_{aer}$ that corresponds to both modes, as done in Wu et al., 2016 (complete reference is given in the manuscript). This investigation is outside the scope of this paper but we mention this as an outlook in the revised manuscript (page 7 lines 185-191).

**[7] P8 L212 "*the error analysis follows a two-step approach*"**
**Could you briefly explain the reasoning for splitting it into two steps?**
In the linear error analysis, employing a one-step or a two-step approach would lead to the same results. However, with the two-step approach it is easier to understand how the aerosol errors are included in the XCO2 error computation, as explicitly expressed in equation 19.
We added a sentence in the manuscript (section 3.2 page 9 lines 230-231) to make this point clear.

**[8] Figure 1 and corresponding text: make it clear when you are talking about the true scene properties vs. the retrieved properties.**
Noted. The horizontal axis labels of Figures 1 and also 7 have been adjusted.

**[9] P14 L357 "*With a $P_{SD}$ above 2 ppm, $XCO_2$ retrievals based on only spectrometer measurements do not meet the mission requirements by a very wide margin (note that we do not apply postretrieval filtering here).*"**
**Can you comment on how much you might expect $P_{SD}$ to be reduced with post-filtering in these simulations?**

When we filter out the converged runs with retrieved aerosol optical depth (at 765 nm) > 0.3, the $P_{SD}$ reduces to 1.95 ppm. Lowering the AOD threshold to 0.2 reduces $P_{SD}$ only slightly to 1.66 ppm.

This information has been added to the manuscript (section 5 page 15 lines 376-379).

**[10] P17 L415 *"Given that the improvement in ΔI=I from 3% to 2% is a major technical challenge..."***
**Would reducing ΔDLP to 0.002 be easier?**
We know that ΔDLP of 0.003 is feasible based on the SPEXone heritage, but reducing ΔDLP to 0.002 would be a major technical challenge as well.

**[11] Figure 5: why does case 1 benefit so much from the SWIR bands?**
A possible explanation could be that for case 1 the additional SWIR bands allow for better characterization of the directional and polarization surface properties, which are  spectrally neutral. Given the relatively small contribution of the case-1 fine mode aerosol to the SWIR bands, the surface can be well separated from the aerosol contribution. The better surface characterization in turn leads to a more accurate aerosol and XCO2 retrieval.

**[12] P21 L465 *"The details of our proposed baseline setup for the MAP-band concept is provided in Table 8."***
**Maybe put this line earlier in Section 6.3.**
We have moved that line to the end of the preceding paragraph (page 21 line 485).

**[13] P25 L537 *"The baseline setups of MAP-mod and MAP-band have generally similar performance."***
**But do the few scenarios where the MAP-band instrument does really poorly mean that MAP- mod is the better choice? Briefly discuss if you recommend one MAP instrument over the other, or if either would be acceptable.**
We performed joint retrievals on the test ensemble with the MAP-band concept and included the analysis in section 7. Based on the results, joint retrievals with either MAP-mod or MAP-band fulfil the CO2M requirements. In this context, either concept is acceptable.

**Technical comments:**
**[14] P2 L50: change to "shorten or lengthen"**

Corrected (now line 52).

**[15] P3 L68: change to "It clearly shows the benefit..."**
We cannot identify the relevant text on the indicated page and line.

**[16] P5 L135 change to "spectrometer"**
Corrected (now line 145).

**[17] P16 L406 looks like the highest value in Fig. 2 is actually just above 2.5, not ~2.4.**

Yes, it is actually 2.52 ppm. The text now (page 17 line 432) reads: "<$\Delta$XCO2> can be as high as 2.52 ppm for the highest $\Delta$I/I and $\Delta$DLP considered here."

**[18] P24 L511 change to "custom"**

Corrected (now page 26 line 554).

**[19] P26 L549 remove "to a large extent"**

Removed (now page 27 line 593).

---

## Author Comment (AC3)

**Reply to Annmarie Eldering**

We thank the referee for her time and effort in critically reading and reviewing our manuscript. Below we reproduce the questions/comments in bold and address them in plain text. Changes in the manuscript are marked in different colors. In response to the overall reviews, we have expanded the manuscript to include

- joint retrieval results in which we adopt the MAP-band concept (section 7), and
- discussion on the retrieved aerosol properties in Appendix B.

**General comments:**

This is a very interesting paper that addresses a very important, relevant question. The authors use a simulation system to investigate the benefits of including multiple angles and polarization sensitivity for instrumentation aimed at characterizing XCO2. It is a well structure paper - clearly explains the experiment that was conducted and the outcome. There are a few areas where I'd like to see some additional information, as this work suggests a real transformation in our ability to retrieve XCO2 from space, so any critical assumptions should be articulated.

Thank you for your positive remarks and careful reading.

**Specific comments:**

[1] The performance shown in Figure 7 is astonishingly good. It would be a significant advancement if we can see this performance with remote sens- ing data. Therefore, it is critical that there is an understanding of how the simulation relates to actual data. Without experimental data that has the features described, this is hard to do, but I believe that a number of groups have performed simulation and then applied their algorithms to real data - for example RemoteC and ACOS applied to GOSAT and OCO-2? If earlier work informs us how do the simulations compare with the reality we can gain confidence that there are not significant error sources missing from the simulation system. The Wu et al paper on MSR makes a brief comparison between simulations and actual OCO-2 retrievals. I would recommend that the authors review the literature and see if there are more detailed discussions in Butz et al or O'Dell et al. to clarify the expected relationship between simulations of XCO2 retrievals and real life performance.

Studies by Butz et al., Guerlet et al., and Wu et al. (complete references are available in the revised manuscript) indicate that retrieval performance using synthetic data predicts quite well the actual performance using real data from GOSAT or OCO2.

We added some text to address this in the last paragraph of section 7 (page 25 lines 547-549).

**[2] The separation of the spectrometer error out from the other errors is a good strategy. How did you decide on that allocation?**

From our experience with spectrometers and MAP, we consider equal proportions of the error for the two instruments to be reasonable estimates. This error partitioning may change over the course of the mission implementation.

[3] You don't have any discussion of the errors that will come from weaknesses in the forward model - the gas spectroscopy has remained a source of error for the OCO-2 mission, and I fully expect this will remain. How can you also consider that error in your analysis or estimate the impact?

Our analysis does not consider errors in the data product due to spectroscopic uncertainties. In this paper, we define payload and instrument specifications. Spectroscopy is certainly an issue but is of secondary relevance for the work here. We added some text to address this in the last paragraph of section 7 (page 25 lines 545-547).

[4] The Frankenberg et al paper (2012) addresses multiple angle measurements and how they might help both aerosol and xco2 characterization. that paper should be cited in the introduction where a review of literature on the interference of aerosols and the value of multi-angle measurements is presented.

Thank you for pointing this out. The paper is now cited in the third paragraph in the introduction (page 2 line 51, page 3 lines 61-64).

[5] Is the the linear error analysis section with all the OE equations really needed? Citation of earlier papers (such as Hasekamp et al or Kulawik et al) should be sufficient. Alternatively, include the central equation in the paper and the rest in the appendix. Don't need to lay out that math in every paper that uses OE and applied linear error analysis.

We understand your concern. However, the two-step approach in our linear error analysis includes a unique element, i.e. the aerosol contribution to XCO2 errors. Since this new aspect is an integral part of the analysis, we feel that it is important to keep the math in its entirety in the main text.

[6] Is there enough difference in the aerosol variables of the simulation and those in the retrieval?? Some of the terms remain the same, and it isn't clear if that contributes to . I think it would be useful to have one table that has the simulation and retrieval info all in once place. I found myself repeatedly flipping back and forth so I could see how the retrieval set up differed from the simulation set up.

The simulation and the spectrometer-only retrieval adopt different aerosol size distributions. The simulation and the joint retrieval use the same aerosol models (consistent retrieval). Fu & Hasekamp 2018 (the complete reference is available in the manuscript) indicated a limited impact when performing 2-mode aerosol retrievals on synthetic measurements with 5 aerosol modes. Based on this, we consider it appropriate to apply a consistent aerosol model for the purpose of this

paper. Inconsistent joint retrieval is a subject of further investigation and is currently being studied.

Having all retrievals and simulation in one table could potentially be confusing because they are discussed in separate sections. We will take care that Table 2 (joint retrieval state vector) and Table 3 (aerosol set-up in the simulation) are placed on the same page during the final layout editing.

[7] I am very interested in the performance of the aerosols themselves. The authors just say this won't be addressed. If this is to be written up in a separate paper, say that. If not, some information about the performance should be included. This could even be an appendix - There is a lot of insight to be gained about the overall retrieval system if we see all of the parameters.

We have now added a discussion about the retrieved aerosol properties in the appendix (Appendix B).

[8] What is the variability of water vapor and temperature profile information? Where we they drawn from? Was there any analysis of correlation of errors with these variables?

The water vapor and temperature profiles were drawn from the AFGL atmospheric profiles. This information has been added to the manuscript (page 11 lines 300-302).

We do not analyse the error correlation with water vapor and temperature, which is a typical aspect of spectrometer-only retrievals and has been discussed in the literature. Here we investigate the benefit of the MAP instrument and we focus on the aerosol-induced XCO2 errors. The analysis of the water vapour and temperature interference is therefore outside the scope of the paper.

[9] Again, the results presented here are impressive - a significant advance for remote sensing of XCO2. The simulations are all conducted for land surfaces, as the driver for this work is the study of human emission of CO2. But, for the larger carbon cycle science community, such an advance would be important. Can the authors add a few comments about how this work could be extended for glint measurements or if they explored the performance over water bodies (perhaps at a range of distances from the glint spot)? Or perhaps this is planned work for a future manuscript? For CO2M, coverage over land surfaces will be given the priority so we do not explore water bodies in our analysis. For the glint geometry, the direct light dominates the light path distribution so we expect less atmospheric scattering. Glint-mode performance with the MAP instrument on board CO2M will be the topic of future research.

We have added glint-related remarks to the manuscript (section 7 page 25 lines 541-544).

**Technical corrections:**

**[10] line 33: spelling of Commission Corrected (now line 35).**

[11] line 73: verb and subject don't match. Also, sentence structure us awkward. Suggest rewording to this "Linear error analysis is part of our study, to derive the optimal instrument specification for each of the two MAP concepts with regard to wavelength range, number of viewing angles and the measurement uncertainties."

Suggestion taken (lines 77-80).

- [12] line 77 and following I don't think commas are needed. These sentence are correct if written this way: For the retrieval input we generate synthetic measurements that correspond to an ensemble of atmospheric and geophysical scenes over land. The MAP instrument for which the synthetic measurements are generated is tailored to the CO2M mission precision and accuracy requirements. Corrected (lines 83-87).
- [13] line 382 refer to Equations A1, 2-4. What is A1? There is no appendix that I am aware of.

There is a short Appendix A just after the box-plot figure. It is hopefully easier to find now that the appendix has been expanded to include aerosol analysis.